# Clonal hematopoiesis of indeterminate potential, DNA methylation, and risk for coronary artery disease

M d Mesbah Uddin [1,2], Ngoc Quynh H. Nguyen[3], Bing Yu[3], Jennifer A. Brody [4], Akhil Pampana [1], Tetsushi Nakao [1,2,5,6], Myriam Fornage [7,8], Jan Bressler [3,8], Nona Sotoodehnia[4], Joshua S. Weinstock [9], Michael C. Honigberg [1,2], Daniel Nachun [10], Romit Bhattacharya[1,2,11], Gabriel K. Griffin[12,13,14], Varuna Chander[15,16], Richard A. Gibbs[15,16], Jerome I. Rotter[17], Chunyu Liu[18,19], Andrea A. Baccarelli [20], Daniel I. Chasman [11,21], Eric A. Whitsel [22,23], Douglas P. Kiel [11,24,25,26], Joanne M. Murabito[19,27], Eric Boerwinkle[3,8,15], Benjamin L. Ebert [5,28], Siddhartha Jaiswal [10], James S. Floyd[4,29], Alexander G. Bick [30], Christie M. Ballantyne [31], Bruce M. Psaty [4,29,32], Pradeep Natarajan [1,2,11,34] ✉ & Karen N. Conneely [33,34] ✉

Age-related changes to the genome-wide DNA methylation (DNAm) pattern observed in blood are well-documented. Clonal hematopoiesis of indeterminate potential (CHIP), characterized by the age-related acquisition and expansion of leukemogenic mutations in hematopoietic stem cells (HSCs), is associated with blood cancer and coronary artery disease (CAD). Epigenetic regulators *DNMT3A* and *TET2* are the two most frequently mutated CHIP genes. Here, we present results from an epigenome-wide association study for CHIP in 582 Cardiovascular Health Study (CHS) participants, with replication in 2655 Atherosclerosis Risk in Communities (ARIC) Study participants. We show that *DNMT3A* and *TET2* CHIP have distinct and directionally opposing genome-wide DNAm association patterns consistent with their regulatory roles, albeit both promoting self-renewal of HSCs. Mendelian randomization analyses indicate that a subset of DNAm alterations associated with these two leading CHIP genes may promote the risk for CAD.

Clonal hematopoiesis of indeterminate potential (CHIP) is a common age-related phenomenon in which hematopoietic stem cells (HSCs) acquire leukemogenic mutations resulting in the selection and expansion of a genetically distinct subpopulation of blood cells (variant allele fraction, VAF > 2%)[1]. The prevalence of CHIP detectable through next-generation sequencing of blood DNA is up to 10% of adults >70 years and nearly 20% of adults >90 years[2–4]. CHIP is associated with increased risk for hematological cancers[2], coronary artery disease (CAD)[4,5], congestive heart failure[6], stroke[7], chronic obstructive

pulmonary disease[8–10], osteoporosis[11], and all-cause mortality[2,3]. The genes most commonly mutated in clonal hematopoiesis are the epigenetic regulators *DNMT3A* and *TET2*, and other commonly mutated genes include regulators of HSC proliferation and tumor suppression[2,4].

DNA methylation (DNAm), the chemical addition of a methyl group to DNA at a cytosine followed by a guanosine (CpG), is a commonly studied epigenetic mechanism with important roles in cell and tissue differentiation. Similar to CHIP, DNAm patterns change

distinctly with age, and have been associated with multiple diseases including cancers[12,13] and coronary artery disease[14,15]. Notably, the products of the two most commonly mutated genes in CHIP regulate DNAm, with *DNMT3A* catalyzing de novo methylation, and *TET2* initiating demethylation via conversion of methylated cytosines to 5-hydroxymethylcytosine[16]. During hematopoiesis, HSCs normally acquire DNAm patterns consistent with terminal cell lineage, but knockout of *Dnmt3a* in mice prevents HSCs from establishing new DNAm patterns, leading to a self-renewal pattern[17]. Despite its opposing regulatory role in demethylation, knockdown of *Tet2* led to a similar pattern of increased HSC self-renewal, and global loss of hydroxymethylation in HSCs[18]. These results suggest that both the addition and removal of methyl groups are necessary to promote differentiation of HSCs, and that insight may be gained from examining the relationship between CHIP and DNAm at specific sites across the genome.

We hypothesized that CHIP overall and gene-specific CHIP mutations would be associated with distinct DNAm signatures, given the roles of *DNMT3A* and *TET2* in regulating DNAm. In this study, we conducted multi-ancestry epigenome-wide association meta-analysis of CHIP, followed by enrichment analysis and functional annotation of associated CpG loci, and mediation analysis and Mendelian randomization to examine the potential interplay between CHIP and DNAm in aging and disease.

## Results

### Baseline characteristics of the study population

The characteristics of the discovery cohort CHS ($N = 582$) and replication cohort ARIC ($N = 2655$) study participants are presented in Table 1. In CHS, 61% of participants were female, 48% were African American, and the mean (standard deviation) age was 73.6 (5.2) years at the time of blood draw for whole-genome sequencing (WGS). In ARIC, 61% of participants were female, 71% were African-American, and the mean (standard deviation) age was 57.4 (5.9) years at the time of blood draw for whole exome sequencing (WES). Overall, CHIP prevalence was 14.8% (86/582) in CHS and 5.3% (142/2655) in ARIC. The top three CHIP genes in both cohorts included *DNMT3A*, *TET2* and *ASXL1* (Supplementary Fig. 1a), with median clone sizes in the 0.11–0.27 VAF range (Supplementary Fig. 1b). Among individuals with CHIP, 86% of CHS and 92% of ARIC participants had a single CHIP mutation with VAF > 2% (Supplementary Fig. 1c). CHIP prevalence was 11.48% (21/183) in CHS and 8.14% (72/884) in ARIC at 61–70 years of age (Supplementary Fig. 1d).

### Epigenome-wide association analyses

The EWAS workflow is presented in Supplementary Fig. 2. We performed a multi-ancestry meta-analysis to carry out discovery EWAS in CHS-AA and CHS-EA. We identified 7422, 4528, and 11,805 CpGs that were differentially methylated (FDR < 0.05) in individuals with any CHIP, *DNMT3A* CHIP, and *TET2* CHIP, respectively; 539, 499, and 1595 CpGs were significant according to a Bonferroni criterion ($P < 1.04 \times 10^{-7}$) (Fig. 1a and Supplementary Fig. 3a, b). Among the 478,661 CpGs tested, at FDR < 0.05, the presence of any CHIP was associated with decreasing DNAm at 1.17% (5618) of sites and increasing DNAm at 0.38% (1804) of sites (Fig. 1a, b). Notably, the *DNMT3A* and *TET2* EWAS profiles showed opposing patterns. The presence of *DNMT3A* CHIP was associated with decreasing DNAm at 0.93% (4435) of sites and increasing DNAm at 0.02% (93) of sites (Fig. 1c and Supplementary Fig. 3a). In contrast, the presence of *TET2* CHIP was associated with decreasing DNAm at 0.23% (1092) of sites and increasing DNAm at 2.24% (10,713) of sites (Fig. 1d and Supplementary Fig. 3b). Quantile-quantile plots of expected and observed $-\log_{10}(P)$ are presented in Supplementary Fig. 4a–f. Consistent with the widespread epigenetic regulatory role of the most frequently mutated CHIP genes, the genomic inflation factor was 1.11, 0.92, and 1.45 in any CHIP, *DNMT3A*, and *TET2* CHIP meta EWAS, respectively. In a

## Table. 1 | Baseline Characteristics of the Study Participants

| Characteristic | | Male | | | | | | Female | | | | | |
| --- | --- | --- | --- | --- | --- | --- | --- | --- | --- | --- | --- | --- | --- |
| | | African American | | | European American | | | African American | | | European American | | |
| | | CHS | ARIC | P | CHS | ARIC | P | CHS | ARIC | P | CHS | ARIC | P |
| N | | 103 | 703 | – | 123 | 323 | – | 177 | 1194 | – | 179 | 435 | – |
| Age at time of WGS/WES sample, mean (range), y | | 73.3 (65–88) | 56.8(47–72) | 4E-58 | 73.7 (65–90) | 59.8 (47–71) | 5E-69 | 73.8 (64–91) | 56.4 (47–71) | 2E-109 | 73.6 (65–89) | 59.1 (47–70) | 5E-108 |
| Ever Smoked, n (%) | | 74 (71.8) | 521 (75.3) | 0.47 | 82 (66.7) | 235 (72.8) | 0.22 | 80 (45.2) | 527 (44.5) | 0.86 | 84 (46.9) | 203 (46.8) | 0.93 |
| BMI, mean (SD) | | 26.9 (4.1) | 28.1 (5.0) | 0.0083 | 26.8 (3.6) | 26.7 (3.6) | 0.68 | 29.6 (5.6) | 31.6 (6.7) | 4E-05 | 26.9 (5.3) | 25.8 (5.0) | 0.019 |
| CAD, n (%) | | 1 (1.0) | 53 (7.6) | 3E-06 | 6 (4.9) | 22 (7.0) | 0.39 | 2 (1.1) | 42 (3.6) | 0.011 | 5 (2.8) | 9 (2.1) | 0.63 |
| T2D, n (%) | | 22 (21.4) | 188 (27.1) | 0.19 | 17 (13.8) | 41 (12.7) | 0.77 | 34 (19.2) | 318 (26.9) | 0.018 | 31 (17.3) | 33 (7.6) | 0.0020 |
| CHIP Mutation cases, n (%) | CHIP | 15 (14.6) | 36 (5.1) | 0.0097 | 20 (16.3) | 18 (5.6) | 0.0033 | 26 (14.7) | 59 (4.9) | 5E-04 | 25 (14.0) | 29 (6.7) | 0.011 |
| | Expanded CHIP (VAF > 10%) | 14 (13.6) | 25 (3.6) | 0.0046 | 19 (15.4) | 10 (3.2) | 4E-04 | 22 (12.4) | 31 (2.7) | 1E-04 | 21 (11.7) | 20 (4.7) | 0.0067 |
| | DNMT3A | 6 (5.8) | 25 (3.6) | 0.29 | 8 (6.5) | 14 (4.4) | 0.30 | 15 (8.5) | 46 (3.9) | 0.027 | 6 (3.4) | 20 (4.7) | 0.60 |
| | TET2 | 2 (1.9) | 5 (0.7) | 0.36 | 5 (4.1) | 0 (0.0) | 0.025 | 4 (2.3) | 8 (0.7) | 0.15 | 7 (3.9) | 1 (0.2) | 0.013 |

In CHS, 175 AA and 230 EA participants have 2-timepoint blood DNA methylation samples. P-values are from a two-sided t-test comparing CHS vs. ARIC. AA African ancestry, EA European ancestry, CHS Cardiovascular Health Study, ARIC atherosclerosis risk in communities, BMI body mass index, CAD prevalent coronary artery disease, T2D prevalent type 2 diabetes, CHIP clonal hematopoiesis of indeterminate potential, VAF variant allele fraction.

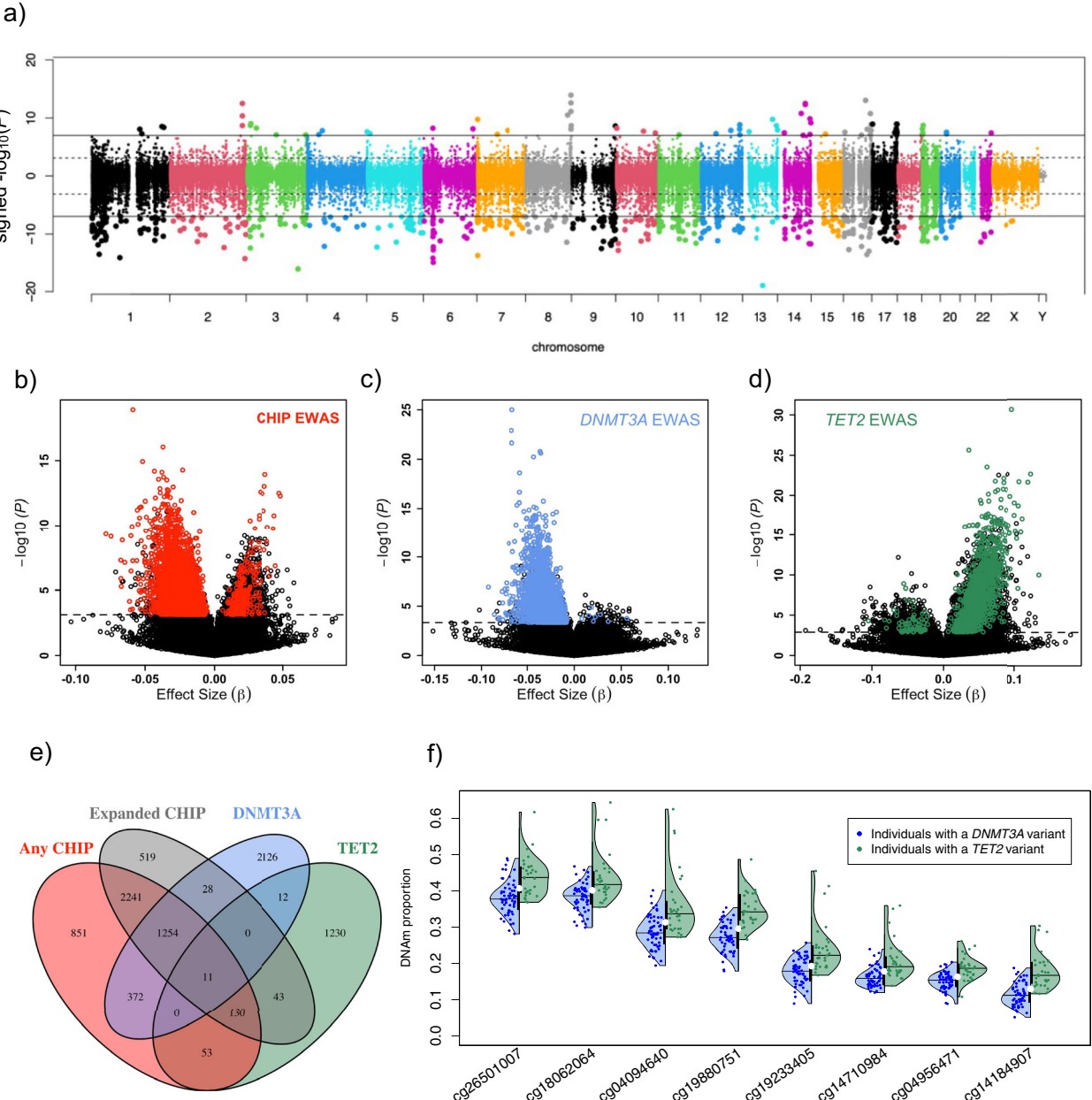

**Fig. 1 | Results from epigenome-wide association studies of four CHIP phenotypes. a** Directional Manhattan plot of discovery multi-ancestry meta-EWAS for any CHIP in CHS cohort, where direction indicates positive vs. negative correlations between CHIP and DNAm. Each dot represents a CpG site, with genomic location on the x-axis and −log₁₀(P)*sign(test statistic) on the y-axis, where P values are based on a two-sided inverse-variance-weighted meta-analysis. Solid horizontal line indicates Bonferroni significance, and dashed line indicates 5% FDR. **b–d** Volcano plots depicting the effect size and −log₁₀(P) from CHS meta EWAS of **b** any CHIP, **c** *DNMT3A* CHIP, and d) *TET2* CHIP. Dashed line indicates FDR < 5%, and colored points highlight CpGs replicated in ARIC cohort. **e** Overlap of replicated CpGs among the four CHIP EWAS. **f** Distribution of DNAm at the eight most significant replicated CpGs associated with both *DNMT3A* and *TET2*. Colored points show DNAm proportions at each CpG for individuals with *DNMT3A* (blue) or *TET2* (green) CHIP, overlaid by density functions for each group and lines representing medians of each distribution. For comparison, medians for individuals without CHIP are shown as white circles.

sensitivity analysis considering a more restricted definition of CHIP requiring larger clone sizes (VAF > 0.10; "expanded CHIP"), results were similar to the EWAS for any CHIP (7881 CpGs associated with an inflation factor of 1.40; Supplementary Fig. 5a–c).

We next performed a replication analysis of the FDR-significant CpGs (FDR < 0.05) in the ARIC study cohorts (1897 AA and 758 EA participants). Approximately 66% (4912/7422), 84% (3803/4528), and 13% (1479/11,801) of CpGs associated with any CHIP, *DNMT3A* CHIP, and *TET2* CHIP were replicated with FDR < 0.05 and concordant effect direction in the multi-ancestry meta-analysis of ARIC-AA and ARIC-EA EWAS. The lower replication rate for *TET2* is likely attributable to the low

prevalence of *TET2* CHIP among ARIC-EA, which only included one individual with *TET2* CHIP (Table 1). When we performed the replication analysis solely in the ARIC-AA cohort, the replication rate was similar, with 1423 of 11,801 CpGs successfully replicating, including 88% (1308) of the sites replicated in the full meta-analysis (Supplementary Data 3). Comparison of our *TET2* discovery results to a previous EWAS of *TET2* CHIP[19] revealed that 63% (6943 out of 11,010 matched CpGs) of CpGs associated with *TET2* CHIP had concordant effect direction and FDR < 0.05 in Tulstrup et al.[19] (Supplementary Data 4), suggesting that our discovery analysis was robust. 1393 of the 1479 CpGs replicated in ARIC were analyzed by;[19] of these, >90% (1258) were corroborated in this

**Table. 2 | Top 20 *DNMT3A*-CHIP-associated CpGs**

| CpG | CHR | Position | Gene[a] | Discovery in CHS | | Replication in ARIC | | Combined meta-analysis | | | | |
|---|---|---|---|---|---|---|---|---|---|---|---|---|
| | | | | β (SE) | P | β (SE) | P | β (SE) | P | Direction[b] | I² | Het P |
| cg04800503 | 17 | 46648533 | *HOXB3* | −0.066 (0.0063) | 9.9E−26 | −0.051 (0.0026) | 1.9E−83 | −0.053 (0.0024) | 3.1E−106 | ---- | 48.1 | 0.1227 |
| cg23014425 | 17 | 46648525 | *HOXB3* | −0.036 (0.0038) | 2.5E−21 | −0.024 (0.0013) | 9.7E−78 | −0.025 (0.0012) | 2.4E−95 | ---- | 80.5 | 0.0015 |
| cg25113462 | 2 | 239299293 | *TRAF3IP1* | −0.036 (0.0038) | 1.6E−21 | −0.056 (0.0033) | 1.8E−66 | −0.048 (0.0025) | 9.5E−83 | ---- | 82.3 | 0.0007 |
| cg07727170 | 15 | 70458214 | | −0.019 (0.0024) | 2.2E−15 | −0.030 (0.0018) | 2.9E−61 | −0.026 (0.0014) | 4.4E−72 | ---- | 77.5 | 0.0039 |
| cg23551720 | 17 | 46633726 | *HOXB3* | −0.033 (0.0040) | 1.8E−16 | −0.050 (0.0035) | 6.2E−47 | −0.043 (0.0026) | 2.2E−59 | ---- | 74.2 | 0.0088 |
| cg03785076 | 2 | 241936915 | *SNED1* | −0.067 (0.0067) | 1.3E−23 | −0.060 (0.0048) | 1.7E−35 | −0.062 (0.0039) | 3.1E−57 | ---- | 39.7 | 0.1737 |
| cg16937168 | 2 | 241936844 | *SNED1* | −0.067 (0.0068) | 2.2E−22 | −0.072 (0.0058) | 2.7E−35 | −0.070 (0.0044) | 7.0E−56 | ---- | 62.9 | 0.0442 |
| cg06186155 | 17 | 46648582 | *HOXB3* | −0.028 (0.0038) | 6.0E−13 | −0.032 (0.0024) | 5.1E−42 | −0.031 (0.0020) | 4.0E−53 | ---- | 0 | 0.6848 |
| cg24400630 | 1 | 89728035 | *GBP5* | −0.044 (0.0069) | 1.1E−10 | −0.049 (0.0035) | 1.2E−42 | −0.048 (0.0032) | 1.2E−51 | ---- | 50.6 | 0.1079 |
| cg23146197 | 12 | 66271002 | *HMGA2* | −0.043 (0.0053) | 6.9E−16 | −0.052 (0.0042) | 4.5E−36 | −0.049 (0.0033) | 6.7E−50 | ---- | 10.6 | 0.3397 |
| cg09749364 | 15 | 40384779 | *BMF* | −0.039 (0.0052) | 9.4E−14 | −0.056 (0.0044) | 7.4E−38 | −0.049 (0.0033) | 1.7E−48 | ---- | 67.2 | 0.0274 |
| cg02836478 | 17 | 46652501 | *HOXB3* | −0.033 (0.0052) | 2.1E−10 | −0.058 (0.0043) | 1.9E−41 | −0.048 (0.0033) | 3.0E−47 | ---- | 80.7 | 0.0014 |
| cg22925751 | 12 | 93509137 | | −0.037 (0.0047) | 3.4E−15 | −0.054 (0.0043) | 8.7E−35 | −0.046 (0.0032) | 5.8E−47 | ---- | 68.2 | 0.0241 |
| cg17839959 | 2 | 178421033 | | −0.021 (0.0028) | 2.6E−14 | −0.023 (0.0019) | 1.8E−32 | −0.022 (0.0016) | 4.1E−45 | ---- | 0 | 0.9771 |
| cg01525376 | 1 | 32716212 | *LCK* | −0.031 (0.0041) | 5.9E−14 | −0.033 (0.0028) | 1.3E−32 | −0.032 (0.0023) | 7.0E−45 | ---- | 20.5 | 0.2869 |
| cg22506548 | 1 | 2996949 | *PRDM16* | −0.065 (0.0107) | 1.1E−09 | −0.043 (0.0034) | 4.6E−37 | −0.045 (0.0032) | 2.8E−44 | ---- | 71.2 | 0.0155 |
| cg25911968 | 2 | 69085916 | | −0.026 (0.0037) | 2.1E−12 | −0.033 (0.0027) | 7.3E−33 | −0.030 (0.0022) | 3.5E−43 | ---- | 0 | 0.4199 |
| cg24771152 | 6 | 31760608 | *VARS* | −0.046 (0.0061) | 2.9E−14 | −0.028 (0.0024) | 1.2E−31 | −0.031 (0.0022) | 1.3E−42 | ---- | 79.8 | 0.0019 |
| cg22946615 | 10 | 30257569 | | −0.036 (0.0064) | 2.0E−08 | −0.042 (0.0033) | 9.7E−36 | −0.040 (0.0030) | 1.8E−42 | ---- | 58.2 | 0.0664 |
| cg13545717 | 9 | 126585875 | *DENND1A* | −0.043 (0.0046) | 5.8E−21 | −0.045 (0.0045) | 9.6E−23 | −0.044 (0.0032) | 4.9E−42 | ---- | 0 | 0.4182 |

[a]Gene annotations reflect those provided in the Illumina manifest file; probes not annotated to a specific gene were left blank.
[b]Direction: + and – indicate positive or negative associations in the CHS AA, CHS EA, ARIC AA, and ARIC EA EWAS, respectively.
I² (heterogeneity statistic) and Het P (heterogeneity P-value) from Cochran's Q test.

comparison, suggesting that our replication results are valid, though conservative due to the low prevalence of *TET2* in the younger ARIC cohort and our stringent FDR-based replication criterion.

Summary statistics for all replicated CpGs from the discovery and replication EWAS, as well as a combined meta-analysis across CHS and ARIC, are presented for the three CHIP categories in Supplementary Data 1–3, and the 20 most significant CpGs from the combined meta-EWAS in *DNMT3A* and *TET2* are shown in Tables 2 and 3. Among replicated sites, 99.8% (3795/3803) of sites associated with *DNMT3A* CHIP showed decreased DNAm, while 94.9% (1404/1479) of sites associated with *TET2* CHIP showed increased DNAm with CHIP. In the combined meta-analysis, the two CpGs most significantly associated with any CHIP and *DNMT3A* CHIP lie within the first intron of *HOXB3*. 86% of replicated CpGs associated with expanded CHIP were also associated with any CHIP (Fig. 1e; Supplementary Fig. 6). However, fewer of the *TET2* and *DNMT3A* CHIP-associated CpGs overlapped with any CHIP and expanded CHIP EWAS, (12–13% and 34–43% respectively for *TET2* and *DNMT3A*). There was limited overlap between *TET2*- and *DNMT3A*-associated CpGs; only 23 CpGs were common between the two, though this was greater than expected by chance (OR = 1.98; $P = 0.003$). Eleven of these CpGs were common among the four CHIP categories (Fig. 1e; Supplementary Fig. 6), where the presence of CHIP was associated with reduced DNAm in all categories for all eleven CpGs (Supplementary Table 1). For the other 12 CpGs, *DNMT3A* CHIP was associated with decreased DNAm while *TET2* CHIP was associated with increased DNAm (Fig. 1f).

### Enrichment analysis
To investigate the regulatory and functional potential of CpG sites associated with CHIP and specifically with *DNMT3A* or *TET2* mutations, we performed a series of analyses to assess whether these sets of CpG sites were enriched relative to other CpGs on the array for regions likely to regulate genes, regions and/or genes associated with specific biological processes, and regions identified as functionally relevant

(via methylation, chromatin accessibility, or gene expression profiles) in HSCs vs. the components of whole blood. We also examined enrichment for genes whose methylation has been found to associate with these mutations in two more extreme contexts: *DNMT3a* knock-out mice[17] and AML patients with driver mutations in *DNMT3A* or *TET2*.

### CHIP-associated CpG sites are enriched in promoter-adjacent regulatory regions
Previous studies have highlighted that the distribution of genome-wide DNAm changes associated with gene regulation and diseases is not random[20]. For example, tissue-specific differentially methylated regions (T-DMR) and cancer-specific DMR (C-DMR) have been found to be depleted in CpG islands (CGI – CpG-rich regions that characterize promoter regions), but 13-fold more frequent in CGI shores ≤2 kb from CGI[21,22]. It has also been reported that methylation shows greater variation and stronger association with nearby gene expression at CGI shores and CGI shelves (adjacent regions 2–4 kb from CGI)[21,23]. To examine the regulatory potential of replicated CpGs, we assessed enrichment for CGI, CGI shores, CGI shelves, and other regions ("open sea"). Replicated CpGs were highly depleted in CGI in all three CHIP categories (0.16 ≤ OR ≤ 0.36; $1.2 \times 10^{-270} \leq P \leq 3.0 \times 10^{-118}$ Supplementary Table 2). CpGs associated with any CHIP or *DNMT3A* CHIP were highly enriched in CGI shores (1.9 ≤ OR ≤ 2.8; $2.5 \times 10^{-267} \leq P \leq 4.0 \times 10^{-78}$), while CpGs associated with *TET2* CHIP were depleted in shores (OR = 0.79; $P = 2.4 \times 10^{-4}$) but enriched in CGI shelves (OR = 1.87; $P = 1.2 \times 10^{-16}$). Sets of CpGs associated with *DNMT3A* or *TET2* CHIP were enriched in open sea regions (1.4 ≤ OR ≤ 2.4, $3.7 \times 10^{-63} \leq P \leq 1.6 \times 10^{-26}$), while CpGs associated with any CHIP were depleted in these regions (OR = 0.82; $P = 8 \times 10^{-11}$).

### CpG sites associated with *DNMT3A* CHIP are enriched in regions associated with stem cell reprogramming and cancer
Because of the role of CHIP mutations in blood cancers and in HSC self-renewal and stemness vs. differentiation, we also examined whether

**Table 3 | Top 20 *TET2*-CHIP-associated CpGs**

| CpG | CHR | Position | Gene[a] | Discovery in CHS | | Replication in ARIC | | Combined Meta-analysis | | | | |
|---|---|---|---|---|---|---|---|---|---|---|---|---|
| | | | | β (SE) | P | β (SE) | P | β (SE) | P | Direction[b] | I² | Het P |
| cg13742400 | 2 | 225639708 | DOCK10 | 0.096 (0.0083) | 2.3E−31 | 0.063 (0.0086) | 3.4E−13 | 0.080 (0.0060) | 3.4E−41 | + + + + | 79.9 | 0.0019 |
| cg17607231 | 2 | 231090329 | SP140 | 0.123 (0.0124) | 2.5E−23 | 0.100 (0.0149) | 1.8E−11 | 0.114 (0.0095) | 6.6E−33 | + + + + | 0 | 0.5287 |
| cg19695507 | 10 | 13526193 | BEND7 | 0.107 (0.0110) | 2.3E−22 | 0.069 (0.0097) | 1.4E−12 | 0.086 (0.0073) | 6.4E−32 | + + + + | 74 | 0.0091 |
| cg26686361 | 16 | 85964073 | | 0.119 (0.0123) | 2.6E−22 | 0.095 (0.0142) | 2.3E−11 | 0.109 (0.0093) | 9.6E−32 | + + + + | 10 | 0.3431 |
| cg12976883 | 2 | 231090376 | SP140 | 0.072 (0.0074) | 1.7E−22 | 0.056 (0.0093) | 1.9E−09 | 0.066 (0.0058) | 5.2E−30 | + + + - | 44.8 | 0.1427 |
| cg13311440 | 1 | 160681404 | CD48 | 0.072 (0.0077) | 3.4E−21 | 0.058 (0.0093) | 5.6E−10 | 0.066 (0.0059) | 2.7E−29 | + + + + | 13.4 | 0.3253 |
| cg10441424 | 5 | 1316636 | | 0.035 (0.0033) | 2.3E−26 | 0.015 (0.0033) | 7.1E−06 | 0.025 (0.0024) | 1.1E−26 | + + + + | 84.1 | 0.0003 |
| cg11887996 | 12 | 120559003 | | 0.061 (0.0060) | 2.9E−24 | 0.032 (0.0071) | 7.6E−06 | 0.049 (0.0046) | 1.7E−26 | + + + − | 75 | 0.0073 |
| cg18098839 | 3 | 167742700 | GOLIM4 | 0.058 (0.0067) | 5.1E−18 | 0.044 (0.0070) | 3.0E−10 | 0.051 (0.0048) | 2.9E−26 | + + + − | 42.6 | 0.1561 |
| cg14064762 | 9 | 123688745 | TRAF1 | 0.068 (0.0080) | 2.0E−17 | 0.063 (0.0100) | 3.6E−10 | 0.066 (0.0062) | 5.0E−26 | + + + + | 0 | 0.8771 |
| cg00476771 | 5 | 64398066 | | 0.094 (0.0101) | 1.5E−20 | 0.073 (0.0143) | 2.9E−07 | 0.087 (0.0083) | 5.0E−26 | + + + + | 0 | 0.6430 |
| cg18642369 | 13 | 99651231 | DOCK9 | 0.111 (0.0130) | 9.4E−18 | 0.104 (0.0176) | 2.7E−09 | 0.109 (0.0104) | 1.7E−25 | + + + + | 0 | 0.4166 |
| cg05165553 | 18 | 77171010 | NFATC1 | 0.076 (0.0080) | 2.1E−21 | 0.042 (0.0088) | 1.5E−06 | 0.061 (0.0059) | 1.0E−24 | + + + − | 82.5 | 0.0007 |
| cg27133780 | 3 | 32474793 | CMTM7 | 0.084 (0.0099) | 2.7E−17 | 0.073 (0.0126) | 5.3E−09 | 0.080 (0.0078) | 1.1E−24 | + + + − | 33.6 | 0.2107 |
| cg20556803 | 7 | 2114593 | MAD1L1 | 0.084 (0.0087) | 2.6E−22 | 0.048 (0.0118) | 5.1E−05 | 0.071 (0.0070) | 1.5E−24 | + + + + | 69.5 | 0.0201 |
| cg08698943 | 10 | 3509758 | | 0.090 (0.0108) | 1.5E−16 | 0.079 (0.0137) | 9.3E−09 | 0.086 (0.0085) | 1.0E−23 | + + + + | 0 | 0.4917 |
| cg08220966 | 10 | 88717364 | MMRN2; SNCG | 0.063 (0.0067) | 7.1E−21 | 0.036 (0.0089) | 5.0E−05 | 0.053 (0.0053) | 3.1E−23 | + + + − | 72.9 | 0.0114 |
| cg09667606 | 6 | 158507930 | SYNJ2 | 0.080 (0.0087) | 3.1E−20 | 0.044 (0.0095) | 3.8E−06 | 0.064 (0.0064) | 3.3E−23 | + + + + | 67.6 | 0.0259 |
| cg13273540 | 3 | 176850227 | TBL1XR1 | 0.081 (0.0096) | 4.4E−17 | 0.056 (0.0101) | 3.6E−08 | 0.069 (0.0070) | 5.1E−23 | + + + + | 60.2 | 0.0564 |
| cg18739367 | 8 | 38330740 | | 0.060 (0.0070) | 5.6E−18 | 0.043 (0.0084) | 4.8E−07 | 0.053 (0.0054) | 5.7E−23 | + + + + | 26.6 | 0.2522 |

[a]Gene annotations reflect those provided in the Illumina manifest file; probes not annotated to a specific gene were left blank.
[b]Direction: + and – indicate positive or negative associations in the CHS AA, CHS EA, ARIC AA, and ARIC EA EWAS, respectively.
I² (heterogeneity statistic) and Het P (heterogeneity P-value) from Cochran's Q test.

CHIP-associated CpGs were enriched for the C-DMR and T-DMR reported in ref. 21 and for induced pluripotent stem cell reprogramming-specific DMR (R-DMR) determined experimentally by Doi et al.[24]. We observed pronounced enrichment in the any CHIP and *DNMT3A* CHIP categories for R-DMR (OR > 3.0; $P < 4.9 \times 10^{-53}$) and C-DMR (OR > 1.6; $P < 2.4 \times 10^{-5}$) (Supplementary Table 3). In contrast, *TET2* CHIP-associated CpGs showed mild but non-significant depletion for both R-DMR and C-DMR (OR ≤ 0.83; $P > 0.05$). All three CHIP categories were significantly depleted for T-DMR, which may reflect that the T-DMR were identified via comparisons of liver, spleen, and brain so do not necessarily vary across blood cell subtypes.

Taken together, the CGI and DMR enrichment analyses suggest distinct regulatory profiles for sets of CpGs associated with any CHIP, *DNMT3A* mutations, and *TET2* mutations. CpGs associated with *DNMT3A* mutations, which tend to be hypomethylated, are more likely to reside in regions associated with gene expression (CGI shores), cancer (C-DMR), and cellular reprogramming (R-DMR). In contrast, CpGs associated with *TET2* mutations tend to be hypermethylated, are enriched in a different set of regions likely to associate with gene expression (CGI shelves), and are not enriched in C-DMR or R-DMR.

**Genes near sites associated with *DNMT3A* and *TET2* CHIP are enriched for distinct biological processes**
Gene ontology (GO) enrichment analysis was performed for genes annotated to replicated CpGs. For the 3803 replicated CpGs associated with *DNMT3A* CHIP, we identified 75 ontologies enriched at FDR < 0.05 and 10 after Bonferroni adjustment for 22,710 ontologies ($P < 2.2 \times 10^{-6}$). A majority of the enriched GO terms were related to developmental and cellular processes, including several terms related to vascular development (Supplementary Data 5). In contrast, among the 1479 replicated CpGs associated with *TET2* CHIP, we identified 27 enriched GO terms at FDR < 0.05 and 9 at Bonferroni significance. Ontologies enriched among *TET2*-associated sites generally related to immune processes, including activation of immune cells of both the

myeloid and lymphoid lineages (Supplementary Data 6). No GO terms were enriched among genes near the 4912 CpGs associated with any CHIP. These results further support a pattern of distinct regulatory consequences associated with *DNMT3A* vs. *TET2* CHIP mutations.

**Sites associated with *DNMT3A* and *TET2* CHIP are enriched for transcription factor binding motifs**
Because DNAm changes may influence gene regulation through modulation of transcription factor binding affinity[25], we used HOMER[26] to investigate enrichment for 364 previously reported transcription factor binding motifs. The 200-bp regions surrounding replicated CpGs associated with *DNMT3A* CHIP were enriched for 40 motifs (FDR < 0.001; Supplementary Fig. 7), including RUNX1 and RUNX2 with roles in HSC and osteoblastic differentiation, five members of the GATA subfamily of transcription factors with roles in development and self-renewal, and five members of the Homeobox family including HOXA9 with roles in AML. Regions surrounding *TET2*-associated sites were enriched for 51 binding site motifs (FDR < 0.001), of which the top 15 belonged to the Erythroblast Transformation Specific (ETS) family of transcription factors with roles in cellular differentiation and pro-liferation (Supplementary Fig. 8). Both *DNMT3A*- and *TET2*-associated sites were highly enriched for motifs for ERG, an essential regulator of hematopoiesis that is aberrantly expressed in leukemia[27,28]. The enrichment of both sets of sites for motifs of transcription factors involved in hematopoiesis and related proliferative processes further supports a functional role for these DNAm changes and their possible involvement in downstream consequences of CHIP such as HSC self-renewal and leukemia.

**Sites associated with *DNMT3A* and *TET2* CHIP have distinct DNAm profiles in HSCs**
Because *DNMT3A* and *TET2* mutations can cause HSCs to propagate through self-renewal rather than differentiate into blood cells[17,18], we examined the DNAm profiles of CpGs associated with *DNMT3A* and

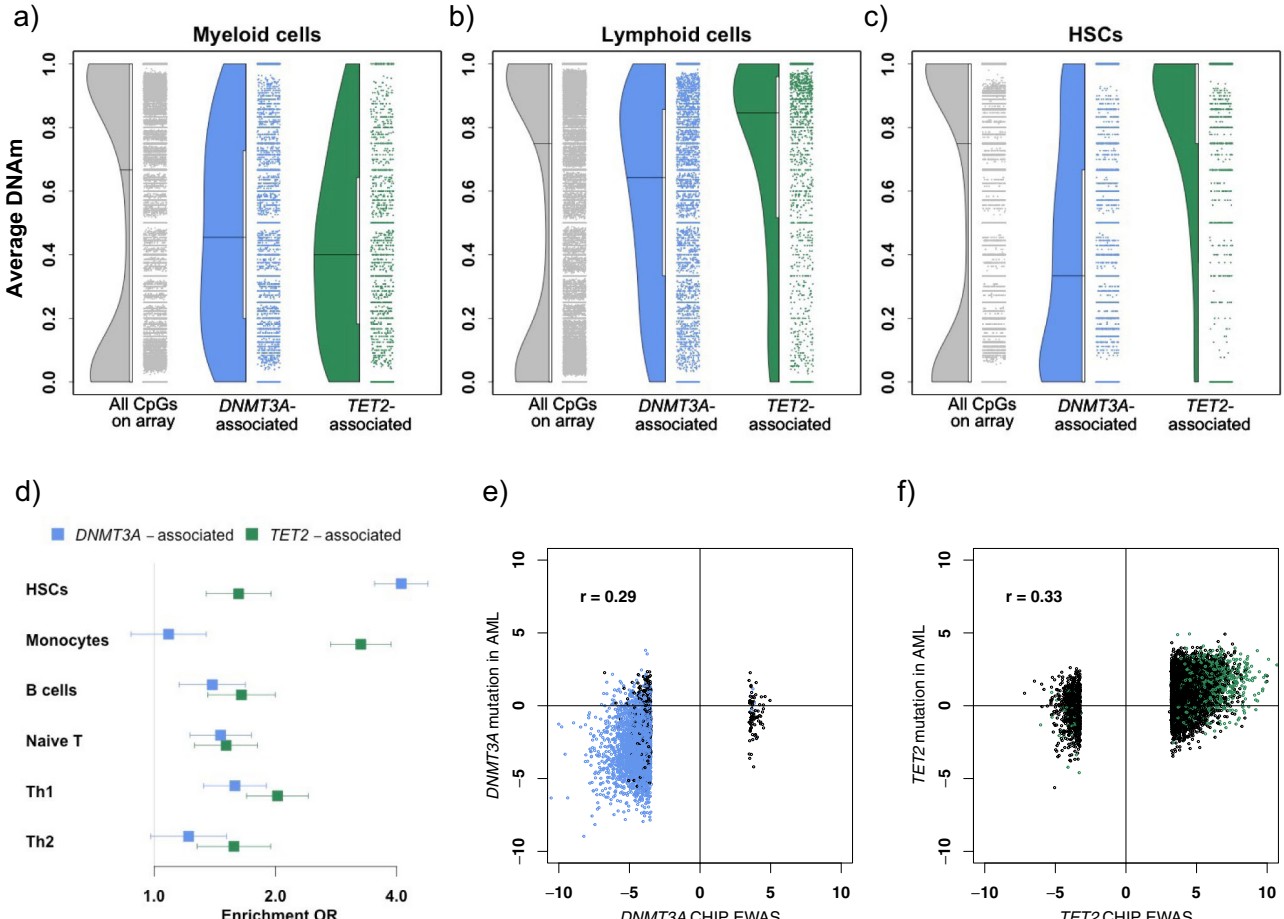

**Fig. 2 | Enrichment patterns among *DNMT3A*- and *TET2*-associated CpGs.**
**a**−**c** Distribution of average methylation levels estimated from external WGBS data for myeloid cells (**a**), lymphoid cells (**b**), and HSCs (**c**) for three sets of CpG sites: all CpGs on Illumina 450 K array (gray), and CpGs showing replicated association with *DNMT3A* CHIP (blue) or *TET2* CHIP (green). Each point represents a CpG, while filled curves show the density function corresponding to all CpGs in each set. Horizontal lines indicate median of distribution. Because of the large number of CpGs considered (*N* = 478,661), all pairwise comparisons between cell types were significant (Two-sided Wilcoxon *P* < 2 × 10⁻¹⁶). **d** Enrichment in cell-specific DHS among the top 1000 *DNMT3A*- or *TET2*-associated CpGs, compared to 1000 random genomic-context-matched CpGs (one-sided binomial test; *N* = 2000). Estimated OR (x-axis,

indicated by filled squares) shows extent to which *DNMT3A*- or *TET2*-associated CpGs are enriched (or depleted) for DHS regions in six distinct cell types (*y*-axis), compared to other sites on the array. Horizontal lines indicate 1−α confidence intervals for estimated OR, using a Bonferroni-adjusted α of 0.05/12. Th1/2: Type 1/ 2 T helper cells. **e**−**f** Comparison of DNAm profiles associated with gene-specific mutations in CHIP vs. AML. Test statistics from EWAS of mutations in *DNMT3A* (**e**) or *TET2* (**f**) in the context of blood samples from healthy individuals with or without CHIP (x-axis; Z-statistics from discovery sample meta-analysis, *N* = 582) vs. tumor samples from patients with AML (*y*-axis; *T*-statistics from EWAS of mutation type in TCGA data, *N* = 127 (**e**) or 108 (**f**)). Black points: FDR < 0.05 in CHS discovery sample but did not replicate; Blue or green points: FDR < 0.05 and replicated in ARIC.

*TET2* CHIP in HSCs vs. downstream blood lineages. Specifically, we compared distributions of average DNAm levels in whole-genome bisulfite sequencing (WGBS) data across myeloid cells, lymphocytes, and HSCs from the BLUEPRINT project[29]—first for the full set of CpGs on the array, and then for sets of CpGs associated with *DNMT3A* and *TET2* mutations. Consistent with previous reports, the distribution of average DNAm proportions across CpGs on the array was bimodal for all three cell types, with the majority of sites either fully methylated or fully unmethylated (gray points in Fig. 2a−c). In contrast, many of the CpGs associated with *DNMT3A* or *TET2* CHIP showed intermediate methylation levels in WGBS data from myeloid cells, with median DNAm proportions of 0.45 and 0.40 among CpGs associated with *DNMT3A* and *TET2* CHIP (Fig. 2a). Both of these sets of CpGs showed higher levels of methylation in lymphoid cells, with median values of 0.64 and 0.85 respectively (Fig. 2b). Notably, in data from HSCs, the two groups of CpGs showed diverging patterns. The majority of sites associated with *TET2* CHIP, which generally showed increased DNAm with CHIP, were fully methylated in HSCs (median = 1.0). In contrast, CpG sites associated with *DNMT3A* CHIP, which generally showed decreased DNAm with CHIP, tended to have lower levels of DNAm in

HSCs (median = 0.33; Fig. 2c). These data suggest that *DNMT3A* and *TET2* CHIP, through opposing mechanisms, each lead to blood DNAm profiles that are more consistent with HSC identity. Because of the large number of CpGs in each group, all pairwise comparisons between cell types were significant (Wilcoxon *P* < 2 × 10⁻¹⁶).

## Sites associated with *DNMT3A* and *TET2* CHIP show differential enrichment for accessible regions in HSCs and progeny cells

We next examined whether CpGs associated with *DNMT3A* and *TET2* CHIP were preferentially located in regulatory regions active in HSCs or downstream blood lineages. Because open chromatin is associated with active regulatory elements and bound transcription factors[30], and demethylation has been shown to induce an open chromatin state[31], we investigated whether sites associated with *DNMT3A* or *TET2* mutations were enriched for accessible regions of chromatin in HSCs and five peripheral blood cell types. Using the eFORGE tool[32], we tested the top 1000 replicated CpG sites associated with *DNMT3A* or *TET2* mutations for enrichment for DNase I hypersensitive (DHS) hotspots identified by ENCODE[33]. Both sets of replicated CpG sites were enriched for DHS hotspots in HSCs, with the enrichment most pronounced

for *DNMT3A*-associated CpGs (OR = 4.1, $P = 1.3 \times 10^{-98}$; Fig. 2d). *TET2*-associated CpGs showed enrichment for DHS hotspots among all five blood cell types (OR > 1.5, $1.1 \times 10^{-63} < P < 2.9 \times 10^{-9}$), with a strong enrichment for regions accessible in monocytes (OR = 3.3; $P = 1.1 \times 10^{-63}$). *DNMT3A*-associated CpGs were enriched for DHS among B cells, naïve T cells, and type 1 T helper cells (1.39 < OR < 1.59, $2.2 \times 10^{-11} < P < 2.4 \times 10^{-6}$) but not monocytes or type 2 T helper cells (OR < 1.22; $P > 0.0042$).

### Genes proximal to *DNMT3A*-CHIP-associated sites show enrichment for HSC marker genes

To further compare the differential DNAm profiles to regulatory profiles in HSCs vs. progeny cells, we tested whether the set of genes annotated to replicated CpG sites associated with *DNMT3A* or *TET2* CHIP mutations showed enrichment or depletion for genesets previously identified as marker genes for HSCs vs. other hematopoietic cells using scRNA-seq data from the Human Cell Atlas bone marrow tissue project[34,35]. Comparing 24 marker genesets, genes near *DNMT3A*-associated sites showed the strongest enrichment for HSC marker genes (OR = 2.6; $P = 2 \times 10^{-25}$), with more modest enrichment for marker genesets for naïve T cells, monocytes, common myeloid progenitor cells, and platelets ($7 \times 10^{-9} < P < 0.0008$; Supplementary Fig. 9). In contrast, genes near *TET2*-associated sites showed nominally significant depletion for HSC marker genes (OR = 0.46; $P = 0.004$), though were enriched for marker genesets for naïve T cells, monocytes, and neutrophils ($1 \times 10^{-14} < P < 0.0008$; Supplementary Fig. 9). Comparison to human orthologs of marker genes identified in murine hematopoietic cells[36] showed a similar enrichment for HSC marker genes among genes proximal to *DNMT3A*-associated CpGs (OR = 2.5; $P = 1.9 \times 10^{-16}$), and a similar (but non-significant) depletion among genes near *TET2*-associated CpGs (OR = 0.71; $P = 0.3$; Supplementary Fig. 10). No other significant enrichments or depletions were observed for genes near *DNMT3A* CHIP-associated sites, though each of the three lymphoid marker genesets showed nominally significant enrichment (0.009 < P < 0.034). Genes near *TET2*-associated sites were enriched for marker genes associated with natural killer cells (OR = 3.7; $P = 2.9 \times 10^{-4}$) and granulocytes (OR = 2.9; $P = 0.0016$), and showed nominally significant enrichment for monocytes (OR = 2.3; $P = 0.0079$). Finally, we evaluated enrichment in a set of 36 orthologous genes (33 encoding transcription factors, and three encoding translational regulators) that were hypothesized as potential HSC reprogramming factors based on >2.5-fold greater expression in murine HSCs compared to 39 other hematopoietic cell types[37]. Genes near *DNMT3A*-associated sites were highly enriched for these 36 factors (OR = 6.6; $P = 5 \times 10^{-39}$), while genes near *TET2*-associated sites were not (OR = 0.57; $P = 0.51$).

Taken together, the results from the cell-type-specific enrichment analyses are consistent with a pattern where the hypomethylation associated with *DNMT3A* mutations occurs in regions associated with an HSC-like epigenetic and transcriptional profile, while the hypermethylation associated with *TET2* mutations occurs primarily in regions associated with accessibility and transcription in differentiated blood cells.

### Genes proximal to both *DNMT3A*- and *TET2*-CHIP-associated sites show enrichment for genes hypo-methylated in *Dnmt3a* knockout mice

Challen et al[17]. previously reported that knockout of *Dnmt3a* in mice is associated with region-specific hypo- and hyper-methylation, and provided lists of genes corresponding to both hyper- and hypo-methylated regions. We assessed whether genes near sites associated with *DNMT3A* or *TET2* CHIP were enriched for human orthologs of these genes. For genes near sites associated with *DNMT3A* CHIP, we observed strong enrichment for orthologs of genes associated with hypo-methylated regions in knockout mice (OR = 2.4; $P = 5 \times 10^{-28}$), but

no enrichment for genes associated with hyper-methylated regions (OR = 0.82; $P = 0.10$). Genes near sites associated with *TET2* CHIP were moderately enriched for orthologs of genes associated with hypo-methylated regions in *DNMT3a* knockout mice (OR = 1.4, $P = 0.014$), but not for genes associated with hyper-methylated regions (OR = 0.80; $P = 0.26$).

### Overlap between replicated sites and sites associated with aging

To examine the extent to which CHIP may contribute to the well-established DNAm signature of aging, we compared the results from our CHS EWAS of CHIP to an EWAS of age performed using the same dataset (see Methods). Of the 4341 sites significantly associated with age ($P < 1.045 \times 10^{-7}$), 243 overlapped with the 7423 sites associated with CHIP in the CHS (OR = 3.86; $P = 7 \times 10^{-64}$), and 176 overlapped with the 4192 sites that replicated in ARIC (OR = 3.95; $P = 3 \times 10^{-46}$), representing greater than-expected overlap in both cases. Comparing the EWAS profiles of CHIP vs. age, there was no correlation between the full set of Z-statistics from the two EWAS ($r = 0.015$), but the CHIP and age EWAS Z-statistics showed substantial correlation when restricting to sites that were significant in the CHIP EWAS ($r = 0.44$) or sites significant in the age EWAS ($r = 0.52$; Supplementary Fig. 11). Among the 4341 sites significant in the age EWAS, we performed Sobel tests[38] of a model where CHIP mediates the relationship between age and DNAm. Nominally significant evidence of mediation ($P < 0.05$) was observed for only 174 of 4341 sites, fewer than the 5% expected by chance. No sites showed significant mediation with FDR < 0.05, suggesting a lack of support for our hypothesis that CHIP could help explain the DNAm signature of aging.

### Overlap between replicated sites and sites associated with leukemogenic mutations in cancer

We next used data generated by TCGA[12] to investigate DNAm profiles in tumor samples from AML patients harboring either *DNMT3A* or *TET2* driver mutations. Of the 396,065 CpGs available for analysis in the TCGA data, 13,031 associated with the presence of a *DNMT3A* mutation in our EWAS of AML patients (see Methods; FDR < 0.05), and 12 associated with the presence of a *TET2* mutation (FDR < 0.05). CpGs that were significant in our replication analysis of *DNMT3A* CHIP were more likely than other CpGs to be significantly associated with a *DNMT3A* mutation in the AML patients (OR = 19.6, $P < 2 \times 10^{-16}$). Sites associated with *TET2* mutations in our replication analysis did not overlap with the 12 CpGs associated with *TET2* mutations in the AML patients but did show greater than-expected overlap with the 26,016 sites nominally associated ($P < 0.05$) with *TET2* mutations in AML (OR = 6.7, $P < 2 \times 10^{-16}$). Figure 2e, f shows that for both genes, differential DNAm was directionally consistent for the CHIP and AML EWAS, with the majority of sites associated with *DNMT3A* CHIP showing decreased DNAm in both contexts and the majority of sites associated with *TET2* CHIP showing increased DNAm in both contexts. This directional consistency led to correlation between *DNMT3A* CHIP and *DNMT3A* AML test statistics ($r = 0.29$; Fig. 2e) and *TET2* CHIP and *TET2* AML test statistics ($r = 0.33$; Fig. 2f); comparable correlations were not observed between *DNMT3A* CHIP test statistics and *TET2* AML test statistics ($r = 0.10$) or vice versa ($r = 0.04$).

### Mendelian randomization analysis of CHIP-associated DNAm and coronary artery disease

To investigate whether DNAm changes may mediate the relationship between CHIP and CAD[4,5], we tested whether DNAm at CHIP-associated CpGs causally influences the risk for CAD using two-sample Mendelian randomization (MR). For this analysis, 2580 CpGs that replicated in at least one CHIP EWAS met the inclusion criteria (5 or more independent associated *cis*-mQTL; see Methods) to be tested for causal association with CAD. The full MR summary statistics are presented in Supplementary Data 7. Genetic instruments were

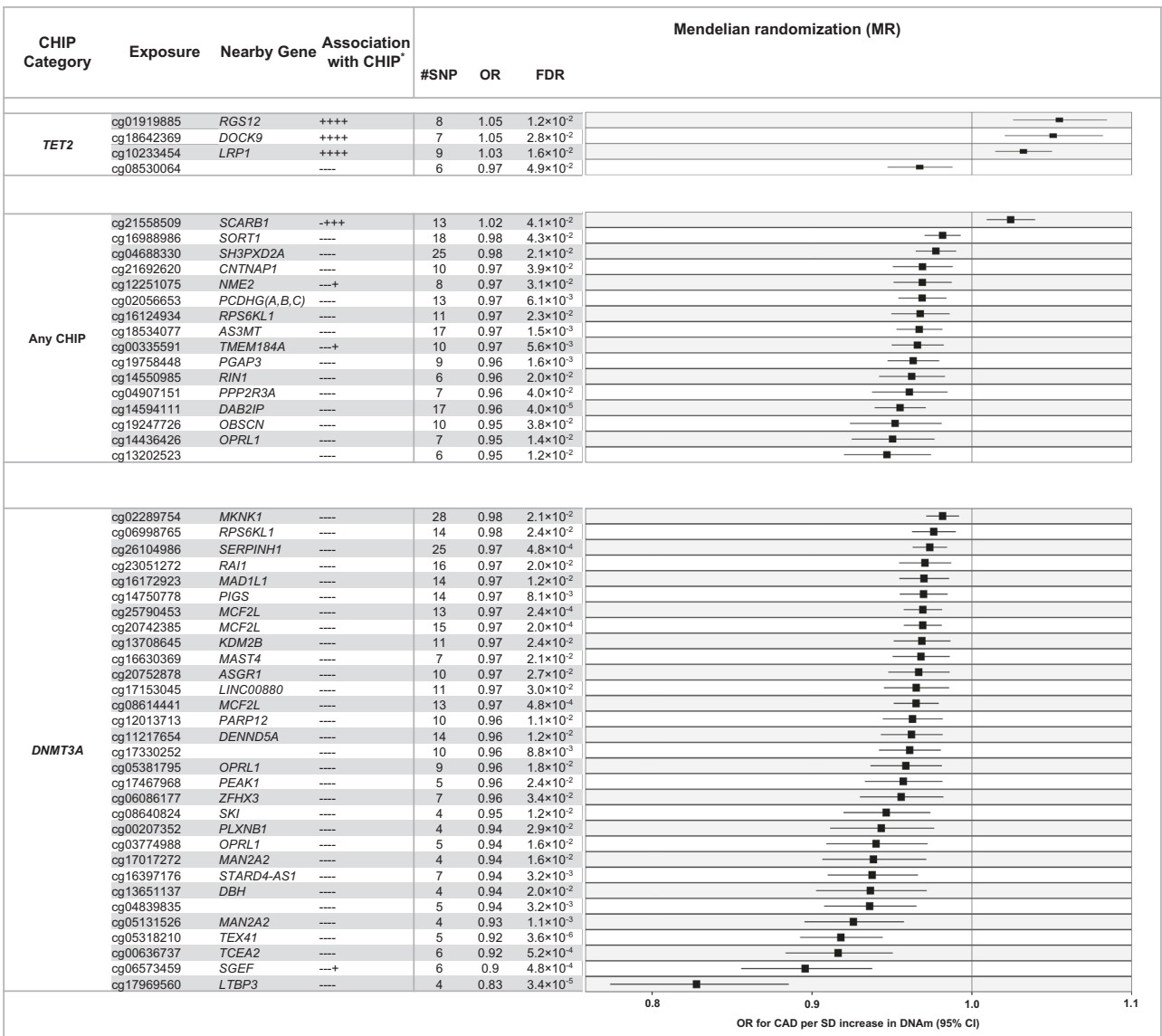

**Fig. 3 | Mendelian randomization analysis of CHIP-associated CpGs and CAD risk.** For sets of replicated CpGs associated with CHIP, *DNMT3A*, and *TET2* (y-axis), the odds ratio (x-axis, indicated by filled squares) reflects the change in CAD risk associated with each SD increase in DNAm, with lines representing 95% confidence intervals estimated by GSMR. GSMR analysis was based on published summary statistics (effect estimates) for cis-mQTL[39] (N = 32,851) and CAD GWAS[41]

(N = 547,261). Only exposure CpGs showing causal evidence in the MR analysis (FDR < 0.05 based on P-values from two-sided χ² test) are presented here; full summary statistics are available in Supplementary Data 7. *"Association with CHIP": "+" or "-" signs indicate effect directions for associations with CHIP, *DNMT3A* or *TET2* in the meta-EWAS of CHS AA, CHS EA, ARIC AA, and ARIC EA EWAS. #SNP: number of SNPs included in the MR analysis for each CpG.

selected as *cis*-mQTL for these CpGs from the GoDMC database[39,39] (http://www.godmc.org.uk/) based on significant SNP-CpG association ($P < 5 \times 10^{-8}$) and partial independence from other SNPs ($r^2 < 0.05$), followed by HEIDI-outlier analysis to remove pleiotropic instruments (see Methods). CAD outcome summary statistics were obtained from the independent meta-analysis of CARDIoGRAMplusC4D[40] and UK Biobank CAD GWAS by van der Harst and Verweij[41]. CHIP has been shown to be associated with increased risk for CAD[3,5,42]. Consistent with the epidemiological observations, 1298 CHIP-associated CpGs were associated with increased risk for CAD in the MR analysis, of which 51 showed significant association with CAD at FDR < 0.05 and 12 at the Bonferroni threshold ($P < 0.05/2580$). However, there were 1282 CHIP-associated CpGs where the change in DNAm was associated with reduced risk for CAD in the MR analysis, of which 53 showed significant association with CAD at FDR < 0.05 and 7 at the Bonferroni threshold. A forest plot representing FDR-significant association between CpGs and increased risk

for CAD is presented in Fig. 3, and scatter plots of corresponding SNP effects on exposures and outcome for the 12 Bonferroni-significant CpG sites were presented in Supplementary Fig. 12. Among the 51 exposure CpGs, 16 were associated with any CHIP, 31 associated with *DNMT3A* CHIP and 4 associated with *TET2* CHIP (Fig. 3). Of the sites associated with *DNMT3A* CHIP, all showed decreased DNAm in individuals with CHIP, and decreased DNAm was associated with increased CAD risk in the MR analysis. For the 4 *TET2*-associated sites, 3 were consistent with CHIP → increased DNAm → increased CAD risk.

Among the *cis*-mQTL used in the MR analyses, several were also *cis*-eQTL in blood[43] (Supplementary Data 8). For example, in CHIP-associated CpG cg14594111, increased DNAm was causally associated with reduced CAD risk (Fig. 3), and the mQTL associated with cg14594111 were also *cis*-eQTL associated with reduced expression of nearby genes including *C5*, *CNTRL*, *GSN*, *PHF19*, and *RAB14* (Supplementary Data 8). Likewise, in *DNMT3A*-associated CpG cg17969560, increased DNAm was causally associated with reduced CAD risk, and

corresponding *cis*-mQTL were also *cis*-eQTL associated with reduced expression of nearby genes, such as *LTBP3* and *NEAT1*. At FDR significance (FDR < 0.05), four *TET2*-associated CpGs showed causal association with CAD where increased DNAm in cg01919885, cg18642369, and cg10233454 were causally associated with increased CAD risk ($1.03 \leq OR \leq 1.05; 1.8 \times 10^{-4} < P < 9.0 \times 10^{-4}$), whereas increased DNAm in cg08530064 was causally associated with reduced CAD risk ($OR = 0.97; P = 2.0 \times 10^{-3}$) (Fig. 3). Here, mQTL alleles associated with increased DNAm were associated with reduced gene expression in *RGS12* (cg01919885), *DOCK9* (cg18642369), *STAT6* (cg10233454) and *TMEM176B/TMEM176A* (cg08530064) (Supplementary Data 8).

## Discussion

Our study identified thousands of CpG sites across the genome whose DNAm was associated with CHIP, including distinct DNAm profiles associated with mutations in each of the two genes most commonly mutated in CHIP. Although this is the first study to identify these opposing methylomic profiles in the context of CHIP, the observed methylomic signatures of *DNMT3A* and *TET2* are consistent with previous work studying mutations in these genes in other contexts. Our observed pattern of decreased DNAm associated with *DNMT3A* mutations and increased DNAm associated with *TET2* mutations is consistent with a recent study of conditional knockout mice that observed a preponderance of hypomethylated regions when comparing regions of open chromatin in *Dnmt3a*-null to control mice, and hypermethylated regions when comparing *Tet2*-null mice to controls[44]. We observed the same patterns of increased or decreased DNAm when we compared AML patients with *DNMT3A* or *TET2* mutations to AML patients with other mutations, and comparison between the AML and CHIP results revealed significant overlap between sets of CpG sites associated with *DNMT3A* or *TET2* CHIP and those associated with mutations in the corresponding gene in AML (Fig. 2e, f). Notably, the DNAm samples used in our study of CHIP were all from healthy participants with no apparent malignancy, and the average VAF was low (~19%, Fig. S1b), compared to the VAF of somatic mutations found in cancer (often 50%). This highlights that aberrant DNAm patterns similar to those found in AML may predate clinical malignancy and can also occur in individuals with CHIP who never progress to cancer. It is also noteworthy that despite the low VAF in most individuals, we were able to observe striking DNAm profiles associated with CHIP, resembling profiles associated with leukemogenic mutations in AML patients or with complete knockout of the genes in mice.

Taken together, these results suggest that there are distinct DNAm profiles associated with impaired activity of *DNMT3A* or *TET2* that can be observed across multiple contexts. Rather than a global gain or loss of DNAm across the genome, each of these DNAm signatures reflects gain or loss of DNAm at specific sites. *DNMT3A*-associated sites showed enrichment for reprogramming-specific DMRs identified by comparing DNAm of fibroblasts to induced pluripotent stem cells derived from those fibroblasts[24], and gene ontology analysis of these signatures identified enrichment for ontologies relevant to developmental and cellular processes among genes located near *DNMT3A*-associated sites, and enrichment for immune processes and immune cell activation among genes located near *TET2*-associated sites. If we consider the canonical regulatory role of DNAm as a silencer of gene expression, this would suggest that the loss of DNAm associated with *DNMT3A* mutations leaves genes active in stem cells free to be expressed, while the gain of DNAm associated with *TET2* mutations silences genes active in the downstream progeny of HSCs.

Along these lines, examination of DNAm levels in HSCs and their downstream progeny revealed that CpG sites associated with *DNMT3A* mutations had decreased DNAm in HSCs compared to myeloid and lymphoid cells, while sites associated with *TET2* mutations showed increased DNAm in HSCs (close to 100% methylation for many sites). *DNMT3A*-associated sites also showed strong enrichment for regions of open chromatin in HSCs, and genes near these sites were enriched for HSC marker genes identified in both humans and mice. The two sites showing the most significant association with *DNMT3A* CHIP mapped to *HOXB3*, a gene found to be overexpressed in acute myeloid leukemia patients with *DNMT3A* mutations[45] and highly expressed in uncommitted hematopoietic cells[46]. In contrast, *TET2*-associated sites were most enriched for regions of open chromatin in monocytes, and genes near these sites showed depletion for HSC marker genes. Overall, these patterns are consistent with a scenario where mutations in either *DNMT3A* or *TET2* both lead to DNAm patterns consistent with HSC-like activity, but through different avenues: *DNMT3A* mutations lead to DNAm loss that upregulates genes related to HSC activity, while *TET2* mutations lead to DNAm gains that downregulate genes related to immune cell activity, thus maintaining an HSC-like state. This scenario aligns well with experimental data showing that knockout of either *Dnmt3a*[17] or *Tet2*[18] results in increased self-renewal of HSCs, but that this occurs through immortalization of HSCs in *Dnmt3a* knockout models[47], while *Tet2* knockout models show normal exhaustion of HSCs but myeloid skewing during differentiation[48]. Our results support models previously suggested for *Dnmt3a* knockout[49] and hypothesized for CHIP in general[50], where DNMT3A loss prevents the silencing of the HSC self-renewal program that normally occurs through methylation of key regions, while TET2 loss prolongs self-renewal by disrupting the differentiation program normally activated via demethylation of key genes and regions.

Notably, EWAS of *DNMT3A* and *TET2* CHIP were recently performed in a smaller set of individuals ($N = 244$), but this study did not identify significant associations between DNAm and *DNMT3A* CHIP[19]. The estimated effect sizes from our *TET2* EWAS showed modest correlation with theirs ($r = 0.269$), and they also noted enrichment for transcription factor motifs from the ETS family among their results for *TET2*, but there was little correlation between effect sizes estimated from their *DNMT3A* EWAS vs. ours ($r = 0.046$). Sample size differences are one possible explanation for the difference between the two studies, but a more likely explanation is that DNAm differences associated with *DNMT3A* CHIP were masked in the previous study due to the inclusion of the top four principal components of DNAm as covariates. Given the striking DNAm profile of *DNMT3A* CHIP we observed in the CHS and ARIC cohorts, and the relatively large prevalence of *DNMT3A* CHIP (55 of 244 individuals in ref. 19) it is likely that both *DNMT3A* CHIP and the cell type proportions (which were included as covariates) were correlated with these principal components, inducing collinearity and masking any association in the previous study. Our high replication rate in ARIC (84% for CpGs significantly associated with *DNMT3A* CHIP in the discovery analysis), along with the alignment of our findings to previously reported experimental results, supports the presence of robust and distinct epigenetic profiles associated with both *DNMT3A* and *TET2* CHIP.

A recent study by Nachun et al.[51] reported associations between CHIP and increased biological age measured by seven different DNAm-based biomarkers of aging. Specifically, the presence of CHIP was associated with an average increase in age acceleration (residual of DNAm-predicted age after adjusting for chronological age) of 1.3–3.1 years across the seven biomarkers. This result supported our initial hypothesis that increased CHIP in older individuals may help explain the genome-wide pattern of age-related DNAm changes. We did observe a moderate correlation in the DNAm profiles associated with age vs. CHIP, but mediation analysis did not provide evidence for CHIP as a potential mediator of the relationship between age and DNAm; however, it may be useful to explore this further in larger studies. Interestingly, Nachun et al. found that stratifying individuals with CHIP based on positive vs. negative age acceleration identified a group at elevated risk for coronary heart disease[51], suggesting that CHIP and DNAm-based age acceleration each contribute independent information about disease risk.

CHIP has been shown to contribute to the increased risk for CAD in older individuals[4,5], but the mechanisms underlying this increased risk are not fully elucidated. Therapeutic hypotheses have focused on inflammasome activation[5,52,53] but the involvement of orthogonal pathways is not well understood. Our MR analysis identified 51 CpG sites where CHIP-associated DNAm changes may contribute to CAD risk. For many of these CpGs, the instrumental variables associated with change in DNAm had an inverse effect on the expression of nearby genes, consistent with the canonical inverse relationship between DNAm and gene expression. Several of these genes have documented functions in lipids metabolism, inflammation, and atherosclerosis. For example, CHIP is associated with reduced DNAm in cg14594111, which is correlated with increased expression of complement *C5*. Increased *C5* level in plasma is correlated with atherosclerotic plaque volume and coronary calcification[54], whereas C5a−a protein fragment of the *C5* protein− promotes atherosclerotic plaque disruptions[55,56]. *DNMT3A* CHIP is associated with reduced DNAm in cg17969560, whose mQTL instruments are correlated with increased expression of *LTBP3* and *NEAT1*. *LTBP3* is implicated in development of aortic aneurysms and dissections[57–59], whereas *NEAT1* is implicated in inflammation and atherosclerosis[60–62]. *TET2* CHIP is associated with increased DNAm in cg10233454, which is correlated with reduced expression of *STAT6*. Lower *STAT6* expression reduces polarization of anti-inflammatory M2 macrophages, increases plaque instability, and thus increases CAD risk[63,64]. *TET2* CHIP is also associated with reduced DNAm in cg08530064, which is correlated with increased expression of *TMEM176A/TMEM176B*. *TMEM176A/TMEM176B* is found to be causally linked with HDL-C metabolism[65,66], and higher expression of *TMEM176B* inhibits the NLRP3 inflammasome by controlling cytosolic $Ca^{2+}$ [67]. NRLP3 inflammasome is involved in atherosclerosis[68] thus higher expression of *TMEM176B/TMEM176A* could have protective CAD effect in individuals with *TET2* CHIP.

Interestingly, the MR analysis also identified 53 CpG sites where CHIP-associated DNAm changes showed a protective effect against CAD. Similar to the 51 "risk" CpG sites, the majority of these sites showed decreased DNAm with CHIP, but for these sites the MR analysis suggested that decreased DNAm at these sites was protective against CAD. Several of these sites were annotated to the first intron or promoter region of *DNMT3B*, which, if upregulated, could potentially help compensate for reduced *DNMT3A* activity. Four were annotated to the first intron of *PRDM16*, which is protective against cardiac hypertrophy and heart failure[69], and whose expression in adipose tissue protects against diet-induced weight gain, likely through greater energy expenditure and activation of brown fat cell (as opposed to white fat cell) activity[70]. While it may seem counterintuitive for CHIP-associated DNAm changes to be identified as protective against CAD, the results of our functional annotation analyses suggest that the primary role of the DNAm changes associated with CHIP is to determine self-renewal vs. differentiation of HSCs. If DNAm does mediate the relationship between CHIP and CAD, it may be that the overall increase in CAD risk is incidental−i.e., that the CHIP-associated DNAm changes include a mix of risk and protective effects that when averaged lead to an increase in risk.

A potential limitation of our study was that DNAm and CHIP were not always measured on concurrent samples. While concurrent measurement in all samples would minimize potential sources of noise, it is important to note that once CHIP is acquired (VAF > 2%), the CHIP clone grows or remains stable in the majority of individuals[71,72]. CHIP was measured either prior to or concurrently with first DNAm measurement for 84% of CHS participants in this study, and prior to or concurrently with the second for >99% of individuals. A second limitation was that CHIP prevalence was lower in our replication sample compared to our discovery sample. This was likely due to the younger age range of the replication sample, as Supplementary Fig. 1d shows comparable prevalence in CHS and ARIC within age groups. Another possible contributing factor is that our discovery vs. replication analyses relied on CHIP called from WGS vs. WES data. However, the previous work[73] has reported similar prevalence for CHIP called via these two approaches, and the prevalence of *DNMT3A* and *TET2* CHIP in the ARIC AA cohort was similar to the population prevalences reported using WGS data for this cohort in[73]. Based on high rates of replication, the differences in prevalence did not appear to hinder our replication of CHIP or *DNMT3A* CHIP. In contrast, only one individual with a *TET2* mutation was present in the ARIC EA cohort studied here. This led to a lower replication rate for CpGs associated with *TET2* CHIP in the multi-ancestry meta-analysis, with only 13% replication of CpGs significant in the *TET2* discovery analysis as compared to 84% replication for *DNMT3A* CHIP and 66% replication for any CHIP. However, comparison to the results from the EWAS of *TET2* CHIP reported in[19] suggested an effective replication rate of 63%, supporting that our discovery results are robust and the lower replication rate stems from low prevalence of *TET2* mutations in the younger ARIC cohort. Future studies in larger and older cohorts will help address this limitation, and will enable the examination of other genes with a lower population prevalence of mutations (e.g. *ASXL1*).

In conclusion, our results are consistent with a pattern where the two most common CHIP mutated genes both promote self-renewal of HSCs through opposing mechanisms, with *DNMT3A* mutations associated with loss of DNAm in regulatory regions near genes associated with HSC activity, and *TET2* mutations associated with gain of DNAm in regulatory regions near genes associated with activity of progeny cells. Mendelian randomization analysis suggests that some of the DNAm alterations associated with CHIP may promote the risk for age-related clinical outcomes such as CAD, while others may be protective against risk.

## Methods

### Study cohorts

The Cardiovascular Health Study (CHS) is a population-based cohort for studying the risk factors for coronary heart disease and stroke in people ≥65 years of age[74]. Our discovery sample consisted of 582 CHS participants who had both CHIP and DNAm data available. DNAm was measured from blood samples taken from these participants in years 5 and 9 ($N = 405$), year 5 ($N = 171$), or year 9 only ($N = 6$). CHIP calls were based on whole-genome sequences (WGS) of blood samples, the majority of which were taken 3 years prior (year 2, $N = 192$) or concurrently (year 5, $N = 294$) with the first DNAm measurement. 86 participants had CHIP calls based on blood samples taken during years 6–9 (so prior to or concurrent with the second DNAm measurement), and the remaining fpur individuals had CHIP calls based on year 10 samples.

Replication samples consisted of 2655 participants from the Atherosclerosis Risk in Communities (ARIC) Study. DNAm was measured from blood DNA samples taken at visit 2 (year 1990–1992; $N = 2228$) and visit 3 (year 1993–1995; $N = 427$). CHIP calls were based on whole exome sequences (WES) of blood DNA samples taken at visit 2 ($N = 2234$) and visit 3 ($N = 421$).

Informed consent was obtained from all study participants, and the study design and methods were approved by the respective institutional review boards at each of the collaborating institutions: University of Washington Institutional Review Board (CHS); University of Mississippi Medical Center Institutional Review Board (ARIC: Jackson Field Center); Wake Forest University Health Sciences Institutional Review Board (ARIC: Forsyth County Field Center); University of Minnesota Institutional Review Board (ARIC: Minnesota Field Center); and Johns Hopkins University School of Public Health Institutional Review Board (ARIC: Washington County Field Center). Each study received institutional certification before depositing sequencing data into dbGaP, ensuring approval by all relevant institutional ethics committees and compliance with relevant ethical regulations.

## DNA methylation measurement

DNA methylation data for CHS and ARIC peripheral blood leukocyte samples were measured via the Illumina Infinium HumanMethylation450 BeadChip (Illumina Inc., San Diego, CA) (see Supplementary Note 1 for details).

## CHIP calls

CHIP was detected previously in CHS from WGS blood DNA in the NHLBI Trans-Omics for Precision Medicine consortium[73]. The same procedure was applied for WES data in ARIC. Mutect2 software[75] was used for somatic mutation calling from WGS data in CHS and WES data in ARIC. CHIP was called from the Annovar[76] annotated VCF files using a custom R script and predefined list of CHIP genes, variants, and rules. The detailed CHIP calling pipeline was previously reported in Bick et al.[73] (https://app.terra.bio/#workspaces/terra-outreach/CHIP-Detection-Mutect2). Individuals with a CHIP mutation at variant allele fraction (VAF) > 2% were defined as CHIP, and those without a CHIP mutation as control. CHIP mutations with VAF > 10% were considered expanded CHIP clones.

## Discovery and replication EWAS

Ancestry-stratified epigenome-wide association analysis was performed using the CpGassoc (v2.60) R package[77]. Separate EWAS were performed in African-American (AA) and European American (EA) individuals within both the discovery and replication cohorts. Each EWAS fit a linear model for each CpG that modeled DNAm proportion as the outcome with CHIP status as the independent variable, adjusted for age, age$^2$, sex, batch, and estimated cell type proportions. In the CHS, individual random effects were included to account for repeated measures from the two longitudinal timepoints. We modeled CHIP status in three different ways, as an indicator variable for the presence of CHIP (yes/no), presence of a CHIP mutation in *DNMT3A*, or the presence of a CHIP mutation in *TET2*. METAL software[78] was used to perform inverse variance weighted fixed effect meta-analysis and Cochran's *Q*-test for heterogeneity[79]. In the discovery analysis, we performed multi-ancestry meta-analyses to combine the results from EWAS within CHS-AA and CHS-EA within each of three EWAS (any CHIP, *DNMT3A* CHIP, and *TET2* CHIP). *P*-values were computed for each site based on two-sided Z-tests, and genome-wide significance was assessed via false discovery rate (Benjamini-Hochberg FDR < 0.05) and Bonferroni threshold $P < 1.04 \times 10^{-7}$ (0.05/478661). CpGs significant in the discovery analysis (FDR < 0.05) were followed up with a replication analysis in ARIC. In the replication analyses, we fit the linear model described above to sets of discovery CpGs from the three aforementioned CHIP categories; models were fit separately in the ARIC-AA and ARIC-EA cohorts, followed by a multi-ancestry meta-analysis. CpGs with FDR < 0.05 and effect direction concordant with the discovery analysis were considered to be successful replications. As a sensitivity analysis, discovery and replication EWAS was also performed for a more restrictive definition of CHIP (VAF > 0.10; Supplementary Note 1).

## Enrichment tests

We tested each set of replicated CpGs from the three EWAS for enrichment of location relative to CpG islands (CGI), previously established differentially methylated regions (DMR), gene ontologies, and transcription factor binding motifs. Within each set of enrichment tests, we used a Bonferroni-adjusted significance criterion to adjust for the three EWAS and the multiple enrichment categories, unless otherwise specified. We used two-sided Fisher's exact tests to test for enrichment of replicated CpGs in relation to CGI, CGI shores (≤2 kb from CGI), shelves (2–4 kb from CGI), and open sea regions (>4 kb from CGI)[20,21]. We used Illumina annotation data on experimentally determined tissue-specific differentially methylated regions (DMR), cancer-specific DMR (CDMR), or reprogramming-specific DMR (RDMR)[24] and performed Fisher's exact tests to elucidate whether

replicated CpGs were enriched in these categories. We performed gene ontology enrichment analysis on sets of genes near replicated CpGs using the missMethyl Bioconductor R package[80] v1.26.1. Finally, we used the HOMER software suite[26] v4.11 to test the 200-bp regions surrounding replicated CpGs for enrichment for previously reported transcription factor binding motifs while accounting for regional differences in GC content. For the HOMER analysis we used the default settings to perform one-sided binomial tests to test for enrichment of known motifs 8, 10, or 12 bp in length, with the 200-bp regions surrounding CpGs not associated with *DNMT3A* or *TET2* mutations (FDR > 0.05) provided as background sequences for comparison.

## Functional annotation of replicated sites

To investigate the functional potential of CpG sites associated with *DNMT3A* or *TET2* mutations, we assessed whether these sets of CpG sites were enriched (relative to other CpGs on the Illumina 450 K array) for regions identified as functionally relevant in the components of whole blood and in HSCs based on cell-type specific DNAm, chromatin accessibility, or gene expression profiles obtained from external reference data. Among genes near CpGs associated with *DNMT3A* or *TET2* CHIP, we also examined enrichment for genes associated with differential methylation in *DNMT3a* knockout mice[17]. For each set of tests, we used a Bonferroni-adjusted significance criterion to adjust for the two sets of CpGs and the multiple enrichment categories, unless otherwise specified.

**Cell-type specific DNAm.** To characterize the DNAm profiles of these CpG sets in HSCs vs. downstream blood lineages, we computed average methylation levels at each CpG according to WGBS data generated as part of the BLUEPRINT project[29]. Preprocessed DNAm data (counts of methylated and total reads by site) were downloaded from GEO series GSE87196 for HSCs and six peripheral blood cell types (CD4+ and CD8+ T cells, B cells, natural killer cells, monocytes, and neutrophils) obtained from purification of blood samples from three healthy donors. To establish average DNAm levels for HSCs vs. progeny cells while maximizing genomic coverage, data were combined across donors and within myeloid (monocytes and neutrophils) and lymphoid (T cells, B cells, and natural killer cells) lineages to form three datasets representing average DNAm levels in myeloid cells, lymphoid cells, and HSCs. The R functions liftover() and findOverlaps() from the rtracklayer (v1.54.0) and GenomicRanges (v1.46.1) Bioconductor packages[81] were used to identify CpGs in the WGBS data that overlapped with CpGs analyzed in the EWAS. Two-sided Wilcoxon tests were then used to compare the cell-type-specific DNAm distributions for our sets of replicated CpGs vs. other CpGs on the array.

**Cell-type specific chromatin accessibility.** To examine whether these CpG sets are enriched for regions of accessible DNA in HSCs and downstream progeny, we used the eFORGE tool[32] v2.0 to test the top 1000 CpG sites in each set for enrichment in regions identified as DNase I hypersensitive (DHS) hotspots generated by the ENCODE project[33] for HSCs and five peripheral blood cell types. eFORGE uses a binomial test to assess whether overlap with DHS hotspots is greater in our sets of replicated CpG sites compared to 1000 genomic-context-matched random probe sets from the same array. To assess significance, we compared the *p*-value from each binomial test to a Bonferroni-adjusted significance criterion adjusted ($\alpha = 0.05/12 = 0.0042$ to account for two sets of CpG sites tested for enrichment in six cell types). For descriptive purposes, we generated odds ratios as the ratio of (1) the odds of sites overlapping DHS in our data to (2) the odds of sites overlapping DHS in the 1000 matched random sets.

**Cell-type specific gene expression.** To examine whether genes proximal to these CpG sites are enriched for cell-type-specific gene expression patterns, we used the Illumina 450 K annotation to

associate each CpG site with a gene, and used two-sided Fisher's exact tests to test these gene sets for enrichment or depletion of sets of genes previously identified as marker genes for HSCs vs. other hematopoietic cells derived from human bone marrow[34] or mouse bone marrow, spleen, and peripheral blood[36,37]. For murine genesets, we identified human orthologs from Ensembl Release 105[82] using the biomaRt Bioconductor package[83]. Enrichment or depletion for marker genesets was assessed using two-sided Fisher's exact test with Bonferroni adjustment for the number of cell types considered.

***Dnmt3a* knockout mice.** We obtained lists of genes previously identified as hypo- or hyper-methylated in *Dnmt3a* knockout mice from the supplemental materials of ref. 17. As above, we identified human orthologs from Ensembl Release 105[82], and tested for enrichment via two-sided Fisher's exact tests.

### Comparison of replicated sites to sites associated with aging

To investigate the overlap between CHIP-associated sites and sites showing differential DNAm with age, we used CpGassoc to perform an EWAS for age in the CHS sample. Similar to our discovery EWAS, this analysis considered DNAm proportion as the outcome, with age as the independent variable and covariates for CHIP, sex, batch, and estimated cell type proportions, and random effects to account for repeated measures. We then compared results from the CHIP vs. age EWAS by considering (1) the proportion of CpGs that were significantly associated with both traits, and (2) Pearson correlation between the meta-analysis Z-statistics from the two EWAS. For CpG sites associated with both traits, we performed two-sided Sobel tests[38] to assess whether CHIP is a potential mediator of the relationship between age and DNAm, defining significance as FDR < 0.05.

### Comparison of replicated sites to sites associated with leukemogenic mutations in cancer

To assess the overlap between sites associated with *DNMT3A* and *TET2* mutations and sites that associate with these mutations in the context of acute myeloid leukemia (AML), we downloaded data generated by The Cancer Genome Atlas (TCGA) that included Illumina 450 K DNAm data for tumor samples from 140 adult AML patients for whom potential driver mutations had been identified via whole-genome or whole-exome sequencing[12]. We then used CpGassoc to perform an EWAS for *DNMT3A* mutations by comparing patients with *DNMT3A* but not *TET2* mutations (N = 28) to patients with other mutations (N = 99), adjusting for age and sex as covariates. We performed a similar EWAS to identify sites with differential DNAm in patients with *TET2* but not *DNMT3A* mutations (N = 9) compared to patients with other mutations (N = 99). We compared the results of these EWAS to the results from our discovery EWAS and meta-analysis, assessing the correlation between test statistics as above.

### Mendelian randomization analysis

To evaluate the potential of DNAm as a potential mediator of the relationship between CHIP and coronary artery disease (CAD), we performed two-sample Mendelian randomization (MR) between exposures (replicated CHIP-associated CpGs) and outcome (CAD). Here, *cis*-methylation quantitative trait loci (*cis*-mQTL) from the GoDMC database[39] were used as instrumental variables (IVs) for the replicated CpGs (excluding MHC region 6: 27486711-33448264) associated with either any CHIP, *DNMT3A* or *TET2* CHIP. The summary statistics of the CAD GWAS meta-analysis of CARDIoGRAMplusC4D[40] and UK Biobank from van der Harst and Verweij[41] were used. We used the generalized summary-data-based Mendelian randomization (GSMR) method of GCTA v1.93.2[84,85] for the analysis.

We prepared a European ancestry LD reference panel using 20,000 random samples from the UK Biobank imputed GWAS dataset. SNPs with allele frequency difference >0.2 between the GWAS

summary dataset (mQTL or CAD QTL) and the LD reference were excluded. In the forward GSMR analysis we considered replicated CpGs with at least five partially independent (linkage disequilibrium, LD $r^2 < 0.05$) *cis*-mQTL with association $P < 5 \times 10^{-8}$. HEIDI-outlier analysis (heterogeneity in dependent instrument, described in Zhu et al.[85]) was then performed to detect and exclude variants with pleiotropic effects and IVs with $P < 0.01$ were excluded. For the FDR-significant (FDR < 0.05) GSMR results, we extracted *cis*-expression quantitative trait loci (*cis*-eQTL; Bonferroni-adjusted $P < 0.05$) from eQTLGen (www.eqtlgen.org)[43] to see whether the *cis*-mQTL used in MR were also *cis*-eQTL, and compared the change in DNAm with corresponding change in gene expression.

### Reporting summary

Further information on research design is available in the Nature Research Reporting Summary linked to this article.

## Data availability

To protect the privacy of research participants and the confidentiality of their data while ensuring that these data are available for appropriate use by researchers, all raw data used in this study are available via controlled access. Individual whole-genome sequencing data for CHS whole genomes generated via TOPMed and the CHIP somatic variant call sets are available through controlled access via dbGaP (https://www.ncbi.nlm.nih.gov/gap/) accession code phs001368. Individual whole exome sequencing data from ARIC are available via dbGaP accession code phs000668. DNA methylation data, as well as phenotypic data, are available via controlled access via ancillary study proposals. Timelines for the approval process range from 4–9 weeks for CHS and 3–6 weeks for ARIC ancillary studies, with specific criteria and proposal forms for the respective studies available at https://chs-nhlbi.org/node/6222 and https://sites.cscc.unc.edu/aric/ancillary-studies-pfg. Summary statistics for replicated associations are available in Supplementary Data 1–4, and full discovery EWAS summary statistics are available from the Downloads page of the Cardiovascular Disease Knowledge Portal (CVDKP; https://cvd.hugeamp.org/downloads.html#other). For enrichment analyses, WGBS data from BLUEPRINT[29] were downloaded from from GEO series GSE87196, murine marker genesets were obtained from Ensembl Release 105[82], and tumor DNAm data from TCGA[12] were downloaded from https://gdc.cancer.gov/about-data/publications/laml_2012.

## Code availability

Code used to generate data presented here is available at: https://github.com/MMesbahU/CHIP-EWAS[86].

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

## Acknowledgements

We gratefully acknowledge funding from the National Institute on Aging (NIA: U34AG051418 to K.N.C., J.M., and A.A.B., R01AG059727 to C.L., T32AG047126, which supported D.N.), the National Institute of Environmental Health Sciences (NIEHS: P30ES009089 to A.A.B.), the National Institute of Arthritis and Musculoskeletal and Skin Diseases (NIAMS: R01AR041398 to D.K.), the National Heart Lung and Blood Institute (NHLBI: DP2-HL157540, to S.J., R01HL148050 to C.B., R01HL141989 to N.S.), and the American Heart Association (940166, 979465, to M.C.H.). P.N. is supported by grants from the NHLBI (R01HL142711, R01HL127564, R01HL148050, R01HL151283, R01HL148565, R01HL135242, R01HL151152), National Institute of Diabetes and Digestive and Kidney Diseases (R01DK125782), Fondation Leducq (TNE-18CVD04), and Massachusetts General Hospital (Paul and Phyllis Fireman Endowed Chair in Vascular Medicine). A.G.B. is supported by NIH grant DP5 OD029586, a Burroughs Wellcome Fund Career Award for Medical Scientists, and a Pew-Stewart Scholar for Cancer Research Award, supported by The Pew Charitable Trusts and the Alexander and Margaret Stewart Trust. R.B. is supported by the John S. LaDue Fellowship in Cardiovascular Disease Research. J.R. is supported in part by the National Center for Advancing Translational Sciences, CTSI grant UL1TR001881, and the National Institute of Diabetes and Digestive and Kidney Disease (NIDDK) Diabetes Research Center (DRC) grant DK063491 to the Southern California Diabetes Endocrinology Research Center, as well as the NIDDK contract R01HL146860. Infrastructure for the CHARGE Consortium is supported in part by the NHLBI grant R01HL105756. The CHS research was supported by NHLBI contracts HHSN268201200036C, HHSN268200800007C, HHSN268201800001C, N01HC55222, N01HC85079, N01HC85080, N01HC85081, N01HC85082, N01HC85083, N01HC85086, 75N92021D00006; and NHLBI grants U01HL080295, R01HL087652,

R01HL103612, R01HL120393, K08HL116640, R01HL092111, R01HL111089, R01HL116747 and U01HL130114 with additional contribution from the National Institute of Neurological Disorders and Stroke (NINDS). Additional support was provided through R01AG023629 from the National Institute on Aging (NIA). Merck Foundation/Society of Epidemiologic Research as well as Laughlin Family, Alpha Phi Foundation, and Locke Charitable Foundation. A full list of principal CHS investigators and institutions can be found at CHS-NHLBI.org. The provision of genotyping data was supported in part by the National Center for Advancing Translational Sciences, CTSI grant UL1TR000124, and the National Institute of Diabetes and Digestive and Kidney Disease Diabetes Research Center (DRC) grant DK063491 to the Southern California Diabetes Endocrinology Research Center. The Atherosclerosis Risk in Communities study has been funded in whole or in part with Federal funds from the National Heart, Lung, and Blood Institute, National Institutes of Health, Department of Health and Human Services, under Contract nos. (HHSN268201700001I, HHSN268201700002I, HHSN268201700003I, HHSN268201700004I, HHSN268201700005I). Funding support for "Building on GWAS for NHLBI-diseases: the U.S. CHARGE consortium" was provided by the NIH through the American Recovery and Reinvestment Act of 2009 (ARRA) (5RC2HL102419). CHARGE sequencing was carried out at the Baylor College of Medicine Human Genome Sequencing Center (U54 HG003273 and R01HL086694). Funding for GO ESP was provided by NHLBI grants RC2 HL-103010 (HeartGO) and exome sequencing was performed through NHLBI grants RC2 HL-102925 (BroadGO) and RC2 HL-102926 (SeattleGO). Funding was also supported by 5RC2HL102419 and R01NS087541. The authors thank the staff and participants of the ARIC study for their important contributions. The content is solely the responsibility of the authors and does not necessarily represent the official views of the National Institutes of Health.

## Author contributions

K.C. and P.N. conceived of the study, with input from D.C., S.J., J.M., A.A.B., D.K., E.W., and C. L, and supervised all analyses. J.Brody, J.Bressler, N.S., M.F, V.C., R.A.G., J.R., E.B., J.F., B.P., and C.M.B. collected, pre-processed, and provided expertise on the data analyzed in this study. P.N., A.G.B., T.N., J.W., D.N., M.H., R.B., and G.G. performed the calling of CHIP in all samples, with input and supervision from B.E. and S.J. M.U. performed the discovery, enrichment, and MR analyses, with input from A.P. and T.N. B.Y. and N.N. performed the replication analyses, and K.C. carried out additional enrichment and functional annotation analyses. M.U. drafted the manuscript with critical input from P.N. and K.C. All authors read, contributed to, and approved the final manuscript.

## Competing interests

B.M.P. serves on the Steering Committee of the Yale Open Data Access Project funded by Johnson & Johnson. P.N. reports grant support from Amgen, Apple, AstraZeneca, Boston Scientific, and Novartis, spousal employment and equity at Vertex, consulting income from Apple, AstraZeneca, Novartis, Genentech / Roche, Blackstone Life Sciences, Foresite Labs, and TenSixteen Bio, and is a scientific advisor board member and shareholder of TenSixteen Bio and geneXwell, all unrelated to this work. J.S.F. has consulted for Shionogi Inc. R.B has consulted for Casana Care Inc, unrelated to this work. J.M. has guest-lectured at Merck, unrelated to this work. M.C.H. has consulted for CRISPR Therapeutics and served on the advisory board for Miga Health, both unrelated to this work. D.K. serves on a DSMB for Agnovos Healthcare, a scientific advisory board for Pfizer and Solarea Bio, and reports grant support from Amgen and Solarea Bio. B.L.E. has received research funding from Celgene, Deerfield, Novartis, and Calico and consulting fees from GRAIL, and is a member of the scientific advisory board and shareholder for Neomorph Inc., TenSixteen Bio, Skyhawk Therapeutics, and Exo Therapeutics. S.J. and A.G.B. are co-founders and equity holders in TenSixteen Bio. The other authors declare no competing interests.

## Additional information

[1]Medical and Population Genetics and Cardiovascular Disease Initiative, Broad Institute of Harvard and MIT, Cambridge, MA 02142, USA. [2]Cardiovascular Research Center, Massachusetts General Hospital, Boston, MA 02114, USA. [3]Department of Epidemiology, Human Genetics, and Environmental Sciences, School of Public Health, The University of Texas Health Science Center at Houston, Houston, TX 77030, USA. [4]Cardiovascular Health Research Unit, Department of Medicine, University of Washington, Seattle, WA 98101, USA. [5]Department of Medical Oncology, Dana-Farber Cancer Institute, Boston, MA 02115, USA. [6]Division of Cardiovascular Medicine, Department of Medicine, Brigham and Women's Hospital, Boston, MA 02115, USA. [7]Brown Foundation Institute of Molecular Medicine, McGovern Medical School, University of Texas Health Science Center at Houston, Houston, TX 77030, USA. [8]Human Genetics Center, School of Public Health, University of Texas Health Science Center at Houston, Houston, TX 77030, USA. [9]Center for Statistical Genetics, Department of Biostatistics, University of Michigan School of Public Health, Ann Arbor, MI 48109, USA. [10]Department of Pathology, Stanford University School of Medicine, Stanford, CA 94305, USA. [11]Department of Medicine, Harvard Medical School, Boston, MA 02115, USA. [12]Department of Pathology, Dana-Farber Cancer Institute, Boston, MA 02215, USA. [13]Department of Pathology, Brigham and Women's Hospital, Boston, MA 02115, USA. [14]Epigenomics Program, Broad Institute of MIT and Harvard, Cambridge, MA 02142, USA. [15]Human Genome Sequencing Center, Baylor College of Medicine, Houston, TX 77030, USA. [16]Department

of Genetics and Genomics, Baylor College of Medicine, Houston, TX 77030, USA. [17]The Institute for Translational Genomics and Population Sciences, Department of Pediatrics, The Lundquist Institute for Biomedical Innovation at Harbor-UCLA Medical Center, Torrance, CA 90502, USA. [18]Department of Biostatistics, School of Public Health, Boston University, Boston, MA 02118, USA. [19]Framingham Heart Study, Boston University and NHLBI/NIH, Framingham, MA 01702, USA. [20]Department of Environmental Health Sciences, Mailman School of Public Health, Columbia University, New York, NY 10032, USA. [21]Division of Preventive Medicine, Brigham and Women's Hospital, Boston, MA 02215, USA. [22]Department of Epidemiology, Gillings School of Global Public Health, University of North Carolina, Chapel Hill, NC 27516, USA. [23]Department of Medicine, School of Medicine, University of North Carolina, Chapel Hill, NC 27516, USA. [24]Hinda and Arthur Marcus Institute for Aging Research, Hebrew SeniorLife, Boston, MA 02131, USA. [25]Department of Medicine, Beth Israel Deaconess Medical Center, Boston, MA 02215, USA. [26]Broad Institute of Harvard and MIT, Cambridge, MA 02142, USA. [27]Department of Medicine, Section of General Internal Medicine, Boston University School of Medicine and Boston Medical Center, Boston, MA 02118, USA. [28]Howard Hughes Medical Institute, Boston, MA 20815, USA. [29]Department of Epidemiology, University of Washington, Seattle, WA 98101, USA. [30]Division of Genetic Medicine, Department of Medicine, Vanderbilt University Medical Center, Nashville, TN, USA. [31]Department of Medicine, Baylor College of Medicine, Houston, TX 77030, USA. [32]Department of Health Systems and Population Health, University of Washington, Seattle, WA 98101, USA. [33]Department of Human Genetics, Emory University School of Medicine, Atlanta, GA 30322, USA. [34]These authors contributed equally: Pradeep Natarajan, Karen N. Conneely. ✉e-mail: pnatarajan@mgh.harvard.edu; kconnee@emory.edu

