## [Peer Review File · Nature Communications]

Clonal hematopoiesis of indeterminate potential, DNA methylation, and risk for coronary artery diseaseEditorial Note: Parts of this Peer Review File have been redacted as indicated to remove third-party material where no permission to publish could be obtained.

REVIEWER COMMENTS

Reviewer #1 (Remarks to the Author):

Uddin et al present a timely analysis of changes in DNA methylation landscapes associated with two of the most common CHIP genes- DNMT3A and TET2. Though these two mutations have both been associated with CHIP and increased incidence of disease, these same phenotypes generated from mutations in these two epigenetic regulation genes have been challenging to interpret as one gene adds methylation (DNMT3) and the other removes methylation (TET2). The authors present a robust number of samples used for a discovery and validation cohort to define epigenetic modifications associated with CHIP mutations and point to epigenetic changes that are enriched for locations near genes expressed in HSCs and progenitors. The authors also tie these locations to reported locations in murine studies, aging, and coronary artery disease. Key locations with DMRs provide insight into targets for evaluating how CHIP can predispose cancer and CAD. The authors have robustly addressed two independent data sets and provide a resource for the community pointing to how CHIP mutations drive the established phenotypes

Minor Comment:

One major strength of this manuscript is the definition of epigenetic changes associated with the CHIP mutations, that are validated in two data sets and in overlap in the murine model as well. These could be further strengthened by more comprehensive analysis of methylation changes that also are seen in then the Tulsrup manuscript (they mention a smaller data set, but overlap for ETS TF binding) but were any of the consistent DMRs also replicated in that date set? Similarly, there are Tet2 mutant mice that could be used to validate the conserved DMRs.

Though beyond the scope of this manuscript, which is already quite robust, validation of the DMRs and expression changes the authors present as progenitor specific could potentially be validated in the murine models.

In the reports of sites with DARs in early progenitor cells that have DMR from the DNMT3A and TET2 mutations, it is unclear of the DHS hotspots are associated with increased or decreased methylation at those sites.

Reviewer #2 (Remarks to the Author):

Uddin et al performed an epigenome-wide association study (EWAS) for clonal hematopoiesis of indeterminate potential (CHIP) in a cohort of 582 Cardiovascular Health Study (CHS) participants, with replication in 2655 Atherosclerosis Risk in Communities (ARIC) Study participants. Two of the most common mutations occur in DNMT3A, which catalyze de novo methylation and in TET2, which initiates demethylation. Therefore, the authors tested the hypothesis that CHIP leads to distinct DNA methylation signatures. CHIP prevalence was 14.8% (86 / 582) in CHS and 5.3% (142 / 2655) in ARIC. The authors first performed EWAS in CHS participants and identified 7,422, 4,528, and 11,805 CpGs that were differentially methylated (FDR<0.05) in individuals with any CHIP, DNMT3A CHIP, and TET2 CHIP, respectively. They then performed a replication analysis and identified 66% (4,912 / 7,422), 84% (3,803 / 4,528), and 13% (1,479 / 11,801) of CpGs associated with any CHIP, DNMT3A CHIP, and TET2 CHIP with FDR<0.05 and concordant effect direction. Enrichment analysis of methylation sites with respect to CpG islands showed distinct categories for replicated CpGs, DNMT3A- and TET2-associated CpGs. The authors then perform a bunch of enrichment analysis as well as Overall, the manuscript deals with an interesting question and some of the analysis performed leads

to interesting findings. However, the rest of the analyses are not well motivated and the implications of their results are not discussed. That issue makes the manuscript read like a bunch of bioinformatics analyses performed just because the authors could do those analyses. Overall, the manuscript lacks a narrative for the data analysis. The authors should either provide a better motivation for some of the analyses and interpret the results in the context of CHIP and disease susceptibility or these analysis results should be removed from the manuscript. Below, I summarize the issues that the authors should address.

- What is the purpose of the enrichment analysis for the location of the CpG islands (pg 8)? It is not clear what biological question is being answered by that analysis. There is also some jargon about shores, open sea regions, etc. I am not sure if they mean anything to those who are not studying DNA methylation patterns. Again, what is really the point of looking for enrichment in cancer specific DMR (CDMR), or reprogramming-specific DMR (RDMR) regions? Either a better rationale should be given for this analysis with appropriate interpretation of the results, or it should be taken out of the manuscript.
- Why did the authors separate the genes annotated to replicated CpGs with either increased or decreased DNA methylation in the CHIP categories? It is possible that a gene that activates a pathway is downregulated and another gene that inhibits a pathway is upregulated; so the net effect on the pathway may be its downregulation but separating the genes into up- and down-regulated categories may miss this overall pathway effect. The authors should perform a combined analysis.
- What is the purpose of the transcription binding site analysis near the differentially methylated CpG islands? What do the authors learn from this analysis that advance our understanding of DNMT3A CHIP's effect on DNA methylation and its consequences for increased disease risk?
- What is the purpose of looking at DNase I hypersensitive site enrichment near replicated CpGs? How is accessible chromatin related to DNA methylation? This context, which can be explained in a sentence or two, is missing from the paper.
- What did the authors learn from their HSC marker gene enrichment analysis about CHIP and DNA methylation?
- What did the authors learn by comparing the human data with the Dnmt3a knock-out mouse data?

Reviewer #3 (Remarks to the Author):

In their manuscript entitled "Clonal hematopoiesis of indeterminate potential, DNA methylation, and risk for coronary artery disease" Uddin and colleagues investigate the effects of clonal hematopoiesis of indeterminate potential (CHIP) on DNA methylation. They show that the two most frequently mutated genes, DNMT3A and TET2, have opposite effects on DNA methylation in (CHIP) and that some of the changes in DNA methylation are causally linked to cardiovascular disease. Overall, the manuscript fails to convey the feeling that this is a unified body of work. The replication of the TET2 data is based on a single case and it should be removed. Perhaps it would be a better fit as a letter/short communication on a more specialized journal.

Main criticisms:

Is figure 1 necessary or would it be better as supplementary item?

The presence of a single TET2 case in the replication cohort should be made clear and the validation downplayed or removed altogether.

The enrichment analyses read like a long and disjointed list.

Minor issues

The first sentence of the abstract sounds awkward.

The last sentence of the first paragraph of the introduction is not referenced.

Table 1 should be split also by sex and the different variables tested between populations and sexes.

The frequencies of CHIP reported seems at odds with ones in the cited literature, the authors should explain why.

Figure 2e would be easier to read as an UpSet plot.

There are some sentences in the result section that would be better placed in the discussion (HOXB3).

The authors should be careful in using differences in p-value to infer strength of association when comparing different sets of CpG islands.

In the discussion there is no mention of the Mendelian Randomization results with opposite effect directions for associations with CHIP, DNMT3A or TET2 in the meta-EWAS of CHS AA, CHS EA, ARIC AA and ARIC EA EWAS.

Response to Reviewers

We thank the reviewers for their thoughtful comments and criticism of the manuscript. We addressed all issues raised and provided a point-by-point response to all comments. Key changes to the manuscript are reprinted below where relevant and highlighted in yellow in the manuscript.

Reviewer #1 (Remarks to the Author):

Uddin et al present a timely analysis of changes in DNA methylation landscapes associated with two of the most common CHIP genes- DNMT3A and TET2. Though these two mutations have both been associated with CHIP and increased incidence of disease, these same phenotypes generated from mutations in these two epigenetic regulation genes have been challenging to interpret as one gene adds methylation (DNMT3) and the other removes methylation (TET2). The authors present a robust number of samples used for a discovery and validation cohort to define epigenetic modifications associated with CHIP mutations and point to epigenetic changes that are enriched for locations near genes expressed in HSCs and progenitors. The authors also tie these locations to reported locations in murine studies, aging, and coronary artery disease. Key locations with DMRs provide insight into targets for evaluating how CHIP can predispose cancer and CAD. The authors have robustly addressed two independent data sets and provide a resource for the community pointing to how CHIP mutations drive the established phenotypes

Author Response: We thank the reviewer for their enthusiasm and constructive comments.

Minor Comment:

One major strength of this manuscript is the definition of epigenetic changes associated with the CHIP mutations, that are validated in two data sets and in overlap in the murine model as well. These could be further strengthened by more comprehensive analysis of methylation changes that also are seen in then the Tulstrup manuscript (they mention a smaller data set, but overlap for ETS TF binding) but were any of the consistent DMRs also replicated in that date set?

Author Response: We appreciate this suggestion. We analyzed overlap between FDR significant CpGs from our discovery EWAS with summary statistics from Tulstrup et al 2021, and report the results in the manuscript and in an additional supplementary table. These results do show a higher rate of corroboration (63% showing concordant effect direction and FDR<0.05) than our replication analysis in ARIC, supporting that our discovery analysis is robust, and that our replication results are overly conservative due to the lower prevalence in this younger sample. We have added this analysis to our Results and reference it in Discussion as well.

Manuscript Changes:

In Results: “The lower replication rate for *TET2* is likely attributable to the low prevalence of *TET2* CHIP among ARIC-EA, which only included one individual with *TET2* CHIP (Table 1). When we performed the replication analysis solely in the ARIC-AA cohort, the replication rate

was similar, with 1,423 of 11,801 CpGs successfully replicating, including 88% (1,308) of the sites replicated in the full meta-analysis (Supplementary Table 3). Comparison of our *TET2* discovery results to a previous EWAS of *TET2* CHIP¹⁹ revealed that 63% (6,943 out of 11,010 matched CpGs) of CpGs associated with *TET2* CHIP had concordant effect direction and FDR<0.05 in Tulstrup, et al.¹⁹ (Supplementary Table 4), suggesting that our discovery analysis was robust. 1,393 of the 1,479 CpGs replicated in ARIC were analyzed by¹⁹; of these, >90% (1,258) were corroborated in this comparison, suggesting that our replication results are valid, though conservative due to the low prevalence of *TET2* in the younger ARIC cohort and our stringent FDR-based replication criterion.”

In Discussion (new text in red): “In contrast, only one individual with a *TET2* mutation was present in the ARIC EA cohort studied here. This led to a lower replication rate for CpGs associated with *TET2* CHIP in the multi-ancestry meta-analysis, with only 13% replication of CpGs significant in the *TET2* discovery analysis as compared to 84% replication for *DNMT3A* CHIP and 66% replication for any CHIP. However, comparison to the results from the EWAS of *TET2* CHIP reported in¹⁹ suggested an effective replication rate of 63%, supporting that our discovery results are robust and the lower replication rate stems from low prevalence of *TET2* mutations in the younger ARIC cohort. Future studies in larger and older cohorts will help address this limitation, and will enable the examination of other genes with a lower population prevalence of mutations (e.g. *ASXL1*).”

Similarly, there are Tet2 mutant mice that could be used to validate the conserved DMRs. Though beyond the scope of this manuscript, which is already quite robust, validation of the DMRs and expression changes the authors present as progenitor specific could potentially be validated in the murine models.

Author Response: Thank you for the suggestion, which we agree is interesting, and for noting that functional study in the murine model is beyond the scope of the manuscript. We agree that this is a natural extension of these observations.

In the reports of sites with DARs in early progenitor cells that have DMR from the DNMT3A and TET2 mutations, it is unclear of the DHS hotspots are associated with increased or decreased methylation at those sites.

Author Response: Replicated sites associated with *DNMT3A* mutations are generally (nearly 100%) associated with decreased methylation, and the opposite is true for *TET2*. So the strong enrichment of *DNMT3A*-associated sites among DHS hotspots in HSCs suggests that a pathogenic *DNMT3A* mutation leads to decreased methylation at CpG sites that are overrepresented in regions accessible in HSCs. Similarly, the strong enrichment of *TET2*-associated sites among monocytes suggests that the CpGs showing increased methylation with *TET2* CHIP are preferentially located in regions usually accessible in monocytes (though perhaps made less accessible by the increased methylation). We have added language to clarify this in the manuscript.

Manuscript Changes:

“Among replicated sites, 99.8% (3,795 / 3,803) of sites associated with *DNMT3A* CHIP showed decreased DNAm, while 94.9% (1,404 / 1,479) of sites associated with *TET2* CHIP showed increased DNAm with CHIP.”

“Taken together, the above results are consistent with a pattern where the hypomethylation associated with *DNMT3A* mutations occurs in regions associated with an HSC-like epigenetic and transcriptional profile, while the hypermethylation associated with *TET2* mutations occurs primarily in regions associated with accessibility and transcription in differentiated blood cells.”

Reviewer #2 (Remarks to the Author):

Uddin et al performed an epigenome-wide association study (EWAS) for clonal hematopoiesis of indeterminate potential (CHIP) in a cohort of 582 Cardiovascular Health Study (CHS) participants, with replication in 2655 Atherosclerosis Risk in Communities (ARIC) Study participants. Two of the most common mutations occur in DNMT3A, which catalyze de novo methylation and in TET2, which initiates demethylation. Therefore, the authors tested the hypothesis that CHIP leads to distinct DNA methylation signatures. CHIP prevalence was 14.8% (86 / 582) in CHS and 5.3% (142 / 2655) in ARIC. The authors first performed EWAS in CHS participants and identified 7,422, 4,528, and 11,805 CpGs that were differentially methylated (FDR<0.05) in individuals with any CHIP, DNMT3A CHIP, and TET2 CHIP, respectively. They then performed a replication analysis and identified 66% (4,912 / 7,422), 84% (3,803 / 4,528), and 13% (1,479 / 11,801) of CpGs associated with any CHIP, DNMT3A CHIP, and TET2 CHIP with FDR<0.05 and concordant effect direction. Enrichment analysis of methylation sites with respect to CpG islands showed distinct categories for replicated CpGs, DNMT3A- and TET2-associated CpGs. The authors then perform a bunch of enrichment analysis as well as

Overall, the manuscript deals with an interesting question and some of the analysis performed leads to interesting findings. However, the rest of the analyses are not well motivated and the implications of their results are not discussed. That issue makes the manuscript read like a bunch of bioinformatics analyses performed just because the authors could do those analyses. Overall, the manuscript lacks a narrative for the data analysis. The authors should either provide a better motivation for some of the analyses and interpret the results in the context of CHIP and disease susceptibility or these analysis results should be removed from the manuscript. Below, I summarize the issues that the authors should address.

Author Response: We thank the reviewer for these useful comments. To provide a narrative for our enrichment analyses we now 1) precede the enrichment analyses with a paragraph to explain the rationale and place them all in context, and 2) begin each enrichment sub-section by explaining the rationale for that specific analysis, and 3) provide concluding sentences at key points (every few sub-sections, often beginning with “Taken together”) to summarize what has been learned from each group of related enrichment analyses.

Manuscript Changes:

Paragraph preceding the enrichment analyses “To investigate the regulatory and functional potential of CpG sites associated with CHIP and specifically with *DNMT3A* or *TET2* mutations, we performed a series of analyses to assess whether these sets of CpG sites were enriched (relative to other CpGs on the Illumina 450K array) for regions likely to regulate genes, regions and/or genes associated with specific biological processes, and regions identified as functionally relevant (via methylation, chromatin accessibility, or gene expression profiles) in HSCs vs. the components of whole blood. We also examined enrichment for genes whose methylation has been found to associate with these mutations in two more extreme contexts: *DNMT3a* knockout mice¹⁷ and AML patients with driver mutations in *DNMT3A* or *TET2*.”

Additional interpretation added to Discussion: “This scenario aligns well with experimental data showing that knockout of either *Dnmt3a*¹⁷ or *Tet2*¹⁸ results in increased self-renewal of HSCs, but that this occurs through immortalization of HSCs in *Dnmt3a* knockout models⁴⁷, while *Tet2* knockout models show normal exhaustion of HSCs but myeloid skewing during differentiation⁴⁸. Our results support models previously suggested for *Dnmt3a* knockout⁴⁹ and hypothesized for CHIP in general⁵⁰, where DNMT3A loss prevents the silencing of the HSC self-renewal program that normally occurs through methylation of key regions, while TET2 loss prolongs self-renewal by disrupting the differentiation program normally activated via demethylation of key genes and regions.”

Further changes are shown below in response to specific points raised by the reviewer.

-What is the purpose of the enrichment analysis for the location of the CpG islands (pg 8)? It is not clear what biological question is being answered by that analysis. There is also some jargon about shores, open sea regions, etc. I am not sure if they mean anything to those who are not studying DNA methylation patterns. Again, what is really the point of looking for enrichment in cancer specific DMR (CDMR), or reprogramming-specific DMR (RDMR) regions? Either a better rationale should be given for this analysis with appropriate interpretation of the results, or it should be taken out of the manuscript.

Author Response: Thank you – we have now added rationale for both of these analyses, including additional connective tissue showing how they are related, background information on the shore/shelf designation, an additional informative header describing the main result for C-DMRs and R-DMRs, and a brief paragraph to explain what is learned from these two subsections.

Manuscript Changes:

“Previous studies have highlighted that the distribution of genome-wide DNAm changes associated with gene regulation and diseases is not random²⁰. For example, tissue-specific differentially methylated regions (T-DMR) and cancer-specific DMR (C-DMR) have been found to be depleted in CpG islands (CGI – CpG-rich regions that characterize promoter regions), but 13-fold more frequent in CGI shores ≤ 2 kb from CGI^{21,22}. It has also been reported that methylation shows greater variation and stronger association with nearby gene expression at CGI shores and CGI shelves (adjacent regions 2-4kb from CGI)^{21,23}. To examine the regulatory

potential of replicated CpGs, we assessed enrichment for CGI, CGI shores, CGI shelves, and other regions (“open sea”).”

“Taken together, the CGI and DMR enrichment analyses suggest distinct regulatory profiles for sets of CpGs associated with any CHIP, *DNMT3A* mutations, and *TET2* mutations. CpGs associated with *DNMT3A* mutations, which tend to be hypomethylated, are more likely to reside in regions associated with gene expression (CGI shores), cancer (C-DMR), and cellular reprogramming (R-DMR). In contrast, CpGs associated with *TET2* mutations tend to be hypermethylated, are enriched in a different set of regions likely to associate with gene expression (CGI shelves), and are not enriched in C-DMR or R-DMR. “

-Why did the authors separate the genes annotated to replicated CpGs with either increased or decreased DNA methylation in the CHIP categories? It is possible that a gene that activates a pathway is downregulated and another gene that inhibits a pathway is upregulated; so the net effect on the pathway may be its downregulation but separating the genes into up- and down-regulated categories may miss this overall pathway effect. The authors should perform a combined analysis.

Author Response: We agree with the reviewer’s comments. We re-analyzed the Gene Ontology enrichment using all replicated CpGs (increased or decreased DNAm) and updated the text and Supplementary Table 8 and 9. The combined analysis resulted in 75 GO terms for *DNMT3A* associated CpGs, and 27 GO terms for *TET2* associated CpGs at FDR <5%.

Manuscript Change: “Gene ontology (GO) enrichment analysis was performed for genes annotated to replicated CpGs. For the 3,803 replicated CpGs associated with *DNMT3A* CHIP, we identified 75 ontologies enriched at FDR<0.05 and 10 after Bonferroni adjustment for 22,710 ontologies ($P < 2.2 \times 10^{-6}$). A majority of the enriched GO terms were related to developmental and cellular processes, including several terms related to vascular development (Supplementary Table 8). In contrast, among the 1,479 replicated CpGs associated with *TET2* CHIP, we identified 27 enriched GO terms at FDR<0.05 and 9 at Bonferroni significance (Supplementary Table 9).”

-What is the purpose of the transcription binding site analysis near the differentially methylated CpG islands? What do the authors learn from this analysis that advance our understanding of DNMT3A CHIP’s effect on DNA methylation and its consequences for increased disease risk?

Manuscript Changes:

“Because DNAm changes may influence gene regulation through modulation of transcription factor binding affinity, we used HOMER to investigate enrichment for 364 previously reported transcription factor binding motifs.”

“The enrichment of both sets of sites for motifs of transcription factors involved in hematopoiesis and related proliferative processes further support a functional role for these

DNAm changes and their possible involvement in downstream consequences of CHIP such as HSC self-renewal and leukemia.”

-What is the purpose of looking at DNase I hypersensitive site enrichment near replicated CpGs? How is accessible chromatin related to DNA methylation? This context, which can be explained in a sentence or two, is missing from the paper.

Manuscript Change:

“We next examined whether CpGs associated with *DNMT3A* and *TET2* CHIP were preferentially located in regulatory regions active in HSCs or downstream blood lineages. Because open chromatin is associated with active regulatory elements and bound transcription factors³⁰, and demethylation has been shown to induce an open chromatin state³¹, we investigated whether sites associated with *DNMT3A* or *TET2* mutations were enriched for accessible regions of chromatin in HSCs and five peripheral blood cell types.”

-What did the authors learn from their HSC marker gene enrichment analysis about CHIP and DNA methylation?

Manuscript Change:

“Taken together, the results from the cell type-specific enrichment analyses are consistent with a pattern where the hypomethylation associated with *DNMT3A* mutations occurs in regions associated with an HSC-like epigenetic and transcriptional profile, while the hypermethylation associated with *TET2* mutations occurs primarily in regions associated with accessibility and transcription in differentiated blood cells.”

-What did the authors learn by comparing the human data with the Dnmt3a knock-out mouse data?

Author Response: The purpose was to examine whether the DNAm profiles associated with CHIP overlap with the DNAm profiles associated with these mutations in another context. This is now described in the rationale paragraph, and we summarize our interpretation in the Discussion.

Manuscript Changes:

“We also examined enrichment for genes whose methylation has been found to associate with these mutations in two other contexts: *DNMT3a* knockout mice¹⁷ and AML patients with driver mutations in *DNMT3A* or *TET2*.”

“...the observed methylomic signatures of *DNMT3A* and *TET2* are consistent with previous work studying mutations in these genes in other contexts...It is also noteworthy that despite the low VAF in most individuals, we were able to observe striking DNAm profiles associated with CHIP, resembling profiles associated with leukemogenic mutations in AML patients or with complete knockout of the genes in mice.”

Reviewer #3 (Remarks to the Author):

In their manuscript entitled “Clonal hematopoiesis of indeterminate potential, DNA methylation, and risk for coronary artery disease” Uddin and colleagues investigate the effects of clonal hematopoiesis of indeterminate potential (CHIP) on DNA methylation. They show that the two most frequently mutated genes, DNMT3A and TET2, have opposite effects on DNA methylation in (CHIP) and that some of the changes in DNA methylation are causally linked to cardiovascular disease.

Overall, the manuscript fails to convey the feeling that this is a unified body of work. The replication of the TET2 data is based on a single case and it should be removed.

Perhaps it would be a better fit as a letter/short communication on a more specialized journal.

Author Response: We thank the reviewer for the comments.

Main criticisms:

Is figure 1 necessary or would it be better as supplementary item?

Author Response: We moved Figure 1 to Supplementary Fig.2.

The presence of a single TET2 case in the replication cohort should be made clear and the validation downplayed or removed altogether.

Author Response: There were 14 *TET2* cases in the full replication cohort (Table 1), but 13 of these 14 cases were in the African-ancestry (AA) cohort and we agree that the single *TET2* case in European ancestry ARIC participants (EA) limited our replication success for *TET2* (1479 of 11801 replicated). We now highlight this issue in our Results section and also re-perform the replication with ARIC-EA removed (so in the ARIC-AA cohort only). This sensitivity analysis reveals a similar rate of replication (see “Manuscript Changes” below), suggesting that the results are robust to whether or not we include the ARIC-EA cohort. Following the suggestion of Reviewer 1, we also examined the rate of replication using *TET2* EWAS summary statistics from Tulstrup et al 2021, and report the results in the manuscript and in Supplementary Table 4. These results show a higher rate of corroboration (63% with concordant effect direction and FDR<0.05), supporting that our discovery analysis is robust, and that our replication results are overly conservative due to the lower prevalence in this younger sample. The full set of significant results from both of these new analyses have been added to our results in Supplementary Tables 3 and 4, and we are now clear about this issue both in Results and Discussion.

Manuscript Changes:

In Results: “The lower replication rate for *TET2* is likely attributable to the low prevalence of *TET2* CHIP among ARIC-EA, which only included one individual with *TET2* CHIP (Table 1). When we performed the replication analysis solely in the ARIC-AA cohort, the replication rate was similar, with 1,423 of 11,801 CpGs successfully replicating, including 88% (1,308) of the

sites replicated in the full meta-analysis (Supplementary Table 3). Comparison of our *TET2* discovery results to a previous EWAS of *TET2* CHIP¹⁹ revealed that 63% (6,943 out of 11,010 matched CpGs) of CpGs associated with *TET2* CHIP had concordant effect direction and FDR<0.05 in Tulstrup, et al.¹⁹ (Supplementary Table 4), suggesting that our discovery analysis was robust. 1,393 of the 1,479 CpGs replicated in ARIC were analyzed by¹⁹; of these, >90% (1,258) were corroborated in this comparison, suggesting that our replication results are valid, though conservative due to the low prevalence of *TET2* in the younger ARIC cohort and our stringent FDR-based replication criterion.”

In Discussion: “In contrast, only one individual with a *TET2* mutation was present in the ARIC EA cohort studied here. This led to a lower replication rate for CpGs associated with *TET2* CHIP in the multi-ancestry meta-analysis, with only 13% replication of CpGs significant in the *TET2* discovery analysis as compared to 84% replication for *DNMT3A* CHIP and 66% replication for any CHIP. However, comparison to the results from the EWAS of *TET2* CHIP reported in¹⁹ suggested an effective replication rate of 63%, supporting that our discovery results are robust and the lower replication rate stems from low prevalence of *TET2* mutations in the younger ARIC cohort. Future studies in larger and older cohorts will help address this limitation, and will enable the examination of other genes with a lower population prevalence of mutations (e.g. *ASXL1*).”

The enrichment analyses read like a long and disjointed list.

Author Response: Thank you for this useful feedback: to address this issue and a similar issue raised by another reviewer we have 1) added a paragraph at the beginning of the enrichment analyses to put them in context, and 2) described rationale at the beginning of each sub-section, and 3) added interpretation at key points to summarize what has been learned from each group of related enrichment analyses.

Manuscript Changes: see responses to reviewer 2, who raised a similar issue.

Minor issues

The first sentence of the abstract sounds awkward.

Manuscript Change: We changed “Age-related changes to the epigenome are well-documented, especially the pattern of genome-wide DNA methylation (DNAm) changes observed in blood.” to “Age-related changes to the genome-wide DNA methylation (DNAm) pattern observed in blood are well-documented.”

The last sentence of the first paragraph of the introduction is not referenced.

Manuscript Change: “The genes most commonly mutated in clonal hematopoiesis are the epigenetic regulators *DNMT3A* and *TET2*, and other commonly mutated genes include regulators of HSC proliferation and tumor suppression^{2,4}.”

Table 1 should be split also by sex and the different variables tested between populations

and sexes.

Manuscript Changes: We modified Table 1 to split by sex and provide tests between discovery and replication (reprinted below, with a better formatted version in manuscript).

Characteristic	Male						Female						
	African American			European American			African American			European American			
	CHS	ARIC	P	CHS	ARIC	P	CHS	ARIC	P	CHS	ARIC	P	
N	103	703	-	123	323	-	177	1194	-	179	435	-	
Age at time of WGS/WES sample, mean (Range), y	73.3 (65-88)	56.8 (47-72)	4E-58	73.7 (65-90)	59.8 (47-71)	5E-69	73.8 (64-91)	56.4 (47-71)	2E-109	73.6 (65-89)	59.1 (47-70)	5E-108	
Ever Smoked, n (%)	74 (71.8)	521 (75.3)	0.47	82 (66.7)	235 (72.8)	0.22	80 (45.2)	527 (44.5)	0.86	84 (46.9)	203 (46.8)	0.93	
BMI, mean (SD)	26.9 (4.1)	28.1 (5.0)	0.0083	26.8 (3.6)	26.7 (3.6)	0.68	29.6 (5.6)	31.6 (6.7)	4E-05	26.9 (5.3)	25.8 (5.0)	0.019	
CAD, n (%)	1 (1.0)	53 (7.6)	3E-06	6 (4.9)	22 (7.0)	0.39	2 (1.1)	42 (3.6)	0.011	5 (2.8)	9 (2.1)	0.63	
T2D, n (%)	22 (21.4)	188 (27.1)	0.19	17 (13.8)	41 (12.7)	0.77	34 (19.2)	318 (26.9)	0.018	31 (17.3)	33 (7.6)	0.0020	
CHIP Mutation cases, n (%)	CHIP	15 (14.6)	36 (5.1)	0.0097	20 (16.3)	18 (5.6)	0.0033	26 (14.7)	59 (4.9)	5E-04	25 (14.0)	29 (6.7)	0.011
	Expanded CHIP (VAF > 10%)	14 (13.6)	25 (3.6)	0.0046	19 (15.4)	10 (3.2)	4E-04	22 (12.4)	31 (2.7)	1E-04	21 (11.7)	20 (4.7)	0.0067
	DNMT3A	6 (5.8)	25 (3.6)	0.29	8 (6.5)	14 (4.4)	0.30	15 (8.5)	46 (3.9)	0.027	6 (3.4)	20 (4.7)	0.60
	TET2	2 (1.9)	5 (0.7)	0.36	5 (4.1)	0 (0.0)	0.025	4 (2.3)	8 (0.7)	0.15	7 (3.9)	1 (0.2)	0.013

The frequencies of CHIP reported seems at odds with ones in the cited literature, the authors should explain why.

Author Response: As reported in the manuscript: "Overall, CHIP prevalence was 14.8% (86 / 582) in CHS and 5.3% (142 / 2655) in ARIC". The average ages of CHS and ARIC participants were ~74, and ~58, respectively, and these prevalences are consistent with reports in the literature for the same age groups (see figure below, taken from reference ⁴ in manuscript).

[redacted]

Figure 2e would be easier to read as an UpSet plot.

Manuscript Change: Thank you, we have now added an UpSet plot as Supplementary figure 6.

There are some sentences in the result section that would be better placed in the discussion (HOXB3).

Manuscript Change: We moved the text describing *HOXB3* to the discussion section.

The authors should be careful in using differences in p-value to infer strength of association when comparing different sets of CpG islands.

Author Response: Thank you for this comment – we agree and have changed our language accordingly wherever applicable, making sure to use odd ratios to describe strength of enrichment (OR>1) or depletion (OR<1), and to use p-values only to assess whether the enrichment is statistically significant after adjustment for multiple testing. This change included the additional calculation of odds ratios for our eFORGE analysis, which is now described in Methods and reported in Results, along with a revised Figure 2d that focuses on odds ratios instead of p-values.

Manuscript Changes:

In Results: “Both sets of replicated CpG sites were enriched for DHS hotspots in HSCs, with the enrichment most pronounced for *DNMT3A*-associated CpGs (OR=4.1, $P=1.3\times 10^{-98}$; Figure 2d). *TET2*-associated CpGs showed enrichment for DHS hotspots among all five blood cell types (OR>1.5, $1.1\times 10^{-63} < P < 2.9\times 10^{-9}$), with a strong enrichment for regions accessible in monocytes

(OR=3.3; $P=1.1\times 10^{-63}$). *DNMT3A*-associated CpGs were enriched for DHS among B cells, naïve T cells, and type 1 T helper cells ($1.39<OR<1.59$, $2.2\times 10^{-11} < P < 2.4\times 10^{-6}$) but not monocytes or type 2 T helper cells ($OR<1.22$; $P > 0.0042$).”

In Methods: “To assess significance we compared the p-value from each binomial test to a Bonferroni-adjusted significance criterion adjusted ($\alpha = 0.05/12 = 0.0042$ to account for two sets of CpG sites tested for enrichment in six cell types). For descriptive purposes, we generated odds ratios as the ratio of 1) the proportion of sites overlapping DHS in our data to 2) the proportion of sites overlapping DHS in the 1000 matched random sets.”

In the discussion there is no mention of the Mendelian Randomization results with opposite effect directions for associations with CHIP, DNMT3A or TET2 in the meta-EWAS of CHS AA, CHS EA, ARIC AA and ARIC EA EWAS.

Author Response: Thank you for bringing up this interesting point; we now address this in Results and Discussion.

Manuscript Changes:

In Results “CHIP has been shown to be associated with increased risk for CAD^{3,5,42}. Consistent with the epidemiological observations, 1298 CHIP-associated CpGs were associated with increased risk for CAD in the MR analysis, of which 51 showed significant association with CAD at $FDR<0.05$ and 12 at the Bonferroni threshold ($P<0.05/2580$). However, there were 1282 CHIP-associated CpGs where the change in DNAm was associated with reduced risk for CAD in the MR

analysis, of which 53 showed significant association with CAD at $FDR < 0.05$ and 7 at the Bonferroni threshold.”

In Discussion: “Interestingly, the MR analysis also identified 53 CpG sites where CHIP-associated DNAm changes were suggested to have a protective effect on CAD. Similar to the 51 “risk” CpG sites, the majority of these sites showed decreased DNAm with CHIP, but for these sites the MR analysis suggested that decreased DNAm at these sites was protective for CAD. Several of these sites were annotated to the first intron or promoter region of *DNMT3B*, which, if upregulated, could potentially help compensate for reduced *DNMT3A* activity. Four were annotated to the first intron of *PRDM16*, which is protective against cardiac hypertrophy and heart failure⁶⁹, and whose expression in adipose tissue protects against diet-induced weight gain, likely through greater energy expenditure and activation of brown fat cell (as opposed to white fat cell) activity⁷⁰. While it may seem counterintuitive for CHIP-associated DNAm changes to be identified as protective for CAD, the results of our functional analyses suggest that the primary role of the DNAm changes associated with CHIP is to determine self-renewal vs. differentiation of HSCs. If DNAm does mediate the relationship between CHIP and CAD, it may be that the overall increase in CAD risk is incidental – i.e., that the CHIP-associated DNAm changes include a mix of risk and protective effects that when averaged lead to an increase in risk.”

REVIEWERS' COMMENTS

Reviewer #3 (Remarks to the Author):

The authors have done a remarkable work and have addressed all my concerns.